# Four-dimensional variational data assimilation with a sea-ice thickness emulator

Charlotte Durand<sup>1</sup>, Tobias Sebastian Finn<sup>1</sup>, Alban Farchi<sup>1,\*</sup>, Marc Bocquet<sup>1</sup>, Julien Brajard<sup>2</sup>, and Laurent Bertino<sup>2</sup>

<sup>1</sup>CEREA, École des Ponts and EDF R&D, Institut Polytechnique de Paris, Île-de-France, France

<sup>2</sup>Nansen Environmental and Remote Sensing Center, 5007 Bergen, Norway

\*Now at European Center for Medium-Range Weather Forecasts, Bonn, Germany

**Correspondence:** Charlotte Durand (charlotte.durand@enpc.fr)

**Abstract.** Developing operational data assimilation systems for sea-ice models is challenging, especially using a variational approach due to the absence of adjoint models. NeXtSIM, a sea-ice model based on a brittle rheology paradigm, enables high-fidelity simulations of sea-ice dynamics at mesoscale resolution ( $\sim 10$  km) but lacks an adjoint. By training a neural network as an Arctic-wide emulator for sea-ice thickness based on mesoscale simulations with neXtSIM, we gain access to an adjoint.

Building on this emulator and its adjoint, we introduce a four-dimensional variational (4D–Var) data assimilation system to correct the emulator’s bias and to better position the marginal ice zone (MIZ). Firstly, we perform twin experiments to demonstrate the capabilities of this 4D–Var system and to evaluate two approximations of the background covariance matrix. These twin experiments demonstrate that the assimilation improves the positioning of the MIZ and enhances the forecast quality, achieving an average reduction in sea-ice thickness root-mean-squared error of 0.8 m compared to the free run. Secondly, we  
assimilate real CS2SMOS satellite retrievals with this system. While the assimilation of these rather smooth retrievals amplifies the loss of small-scale information in our system, it effectively corrects the forecast bias. The forecasts of our 4D–Var system achieve a similar performance as the operational sea-ice forecasting system neXtSIM-F. These results pave the way to the use of deep learning-based emulators for 4D–Var systems to improve sea-ice modeling.

## 1 Introduction

Combining observational data with sea-ice models by data assimilation can improve the accuracy of sea-ice forecasts for practical applications such as maritime routing but is computationally expensive. Deep learning offers a solution by providing efficient neural network emulations that serve as auto-differentiable alternatives to costly physical models, which often have no adjoint available. This enables the implementation of four-dimensional variational data assimilation systems (4D–Var), which can potentially enhance the accuracy and scalability of sea-ice forecasts.  
In this paper, we introduce a 4D–Var system based on the sea-ice thickness (SIT) emulator previously developed by Durand et al. (2024). This emulator aims at reproducing the evolution of the sea-ice thickness as modeled by neXtSIM<sup>the state-of-the-art sea-ice model neXtSIM (Rampal et al., 2016; Ólason et al., 2022). We initially demonstrate the feasibility of employing this emulator and its adjoint within a 4D–Var framework through twin experiments. Subsequently, we present</sup>

promising outcomes derived from applying this approach to real observational data, with the assimilation of SIT retrievals  
from the merged ~~CryoSat-SMOS product~~ product of Cryosat-2 altimeter and the Soil Moisture and Ocean Salinity (SMOS)  
radiometer, called CS2SMOS (Ricker et al., 2017). Forecasts obtained with our system are comparable to those of the operational neXtSIM-F system (Williams et al., 2021) – with -F standing for "Forecast".

Various data assimilation techniques are currently implemented in operational sea-ice forecasting systems (Liu et al., 2019). Ensemble Kalman filters with flow-dependent covariances are predominantly utilized in sea-ice forecasting systems, (Sakov et al., 2012; Kimmritz et al., 2016), particularly for assimilating key variables like sea-ice concentration (Massonnet et al., 2015). Other techniques use covariances that are static in time such as nudging (Lindsay and Zhang, 2006; Tietsche et al., 2013), optimal interpolation (Wang et al., 2013; Ji et al., 2015), and three-dimensional variational assimilation (Hebert et al., 2015; Toyoda et al., 2015; Lemieux et al., 2015) and are employed in diverse systems (Caya et al., 2010; Donlon et al., 2012; Zuo et al., 2019).

Arctic sea-ice thickness is heavily dependent on the sea ice rheological model. The state-of-the-art sea-ice model neXtSIM (Rampal et al., 2016; Ólason et al., 2022) has been developed around brittle rheologies (Girard et al., 2011; Dansereau et al., 2016) to better parameterize the observed small-scale processes of sea ice. Run at a mesoscale resolution of approximately 10 km, the model successfully simulates the observed scaling and multifractal properties of sea ice across space and time (Rampal et al., 2019; Bouchat et al., 2022). Until now, two data assimilation systems have been developed for neXtSIM: an ensemble Kalman filter (Cheng et al., 2023b) has been tested to assimilate observations of the sea-ice thickness and concentration, but for operational forecasting, the neXtSIM-F system resorted instead to a simple nudging technique (Williams et al., 2021).

Various data assimilation techniques are currently implemented in operational sea-ice forecasting systems (Liu et al., 2019). Ensemble Kalman filters with flow-dependent covariances are often utilized in sea-ice forecasting systems, (Sakov et al., 2012; Kimmritz et al., 2016), particularly for assimilating key variables like sea-ice concentration (Massonnet et al., 2015). Other techniques use covariances that are static in time such as optimal interpolation (Wang et al., 2013; Ji et al., 2015), or three-dimensional variational assimilation (Hebert et al., 2015; Toyoda et al., 2015; Lemieux et al., 2015) and are employed in diverse systems (Caya et al., 2010; Donlon et al., 2012). Initialization techniques using nudging (Lindsay and Zhang, 2006; Tietsche et al., 2013) have also been implemented.

In this work, we focus on four-dimensional variational data assimilation methods (4D-Var), which rely on a cost function whose efficient minimization requires gradients and the adjoint of the model as well as a background term that incorporates prior information about the state of the system. The idea behind (strong constraints) four-dimensional variational methods is to estimate a model trajectory that fits the observations at best throughout a time period (Sasaki, 1970; Talagrand and Courtier, 1987), called data assimilation window (DAW). To propagate The propagation of the gradient information from the observational time backwards in time within the DAW, the cost function minimization implicitly depends on the model's adjoint during the cost function minimization. Hence, by updating the analysis at initialization time, the data assimilation accounts for all parts of information up to the end of the DAW.

Adjoints for sea-ice models, including its rheology, are rarely developed because of potential numerical instabilities (Fenty and Heimbach, 2009; Kauker et al., 2009; Fenty and Heimbach, 2013). While adjoints for simplified free-drift models are achievable (Koldunov et al., 2017), they yield limited realism for full Arctic sea-ice simulations. Usui et al. (2016); Toyoda et al. (2015, 2019) de-

veloped an adjoint of their sea-ice model, which relies on an slightly simplified elasto-visco-plastic (EVP) rheology scheme (Hunke and Dukowicz, 1997) within a coupled ocean-sea ice model framework to avoid numerical instabilities. Their approach includes a sensitivity analysis of the adjoint, particularly targeting grid-cells in the Marginal Ice Zone (MIZ) and the central Arctic. Koldunov et al. (2017) incorporated an adjoint based on a viscous-plastic (VP) rheology studied sea-ice concentration assimilation with an adjoint (Hibler, 1979) in the MIT-gcm model. The rheology in those adjoints is either discarded or simplified.

Recent advances in deep learning for sea-ice modeling encompass a range of applications, from the full emulation of variables such as sea-ice thickness (Durand et al., 2024), probabilities of sea-ice coverage (Andersson et al., 2021), or sea-ice concentration itself (Liu et al., 2021), to more specialized tasks like model error correction (Finn et al., 2023) or the emulation of melt ponds (Driscoll et al., 2024). Additionally, techniques to integrate neural networks with data assimilation have been proposed, including learning data assimilation increments for model bias correction (Gregory et al., 2023, 2024a), as well as the calibration of sea-ice forecasts (Palerme et al., 2023).

In the present work, we exploit the sea-ice thickness emulator developed by Durand et al. (2024) for both forward and adjoint modeling, as needed for the 4D–Var data assimilation system. The principle of using an emulator for 4D–Var was introduced by Hatfield et al. (2021) and showcased in Lorenz 63 toy examples (Chennault et al., 2021). Recently, this principle was extended to a Numerical Weather Prediction (NWP) emulator based on ERA5 data (Xiao et al., 2023). However, our study represents the first 4D–Var data assimilation system that is built around an emulator specifically designed to capture the evolution of sea ice and the adjoint of the dynamics.

We explore two setups in this paper. Firstly, we focus on twin data assimilation experiments where observations are artificially generated by introducing noise into neXtSIM simulation outputs. Secondly, we assimilate real satellite observations from the merged CS2SMOS product. The paper is organized as follows. Section 2 presents the sea-ice model , and the additional forcings, and the observations. Section 3 provides a brief description of the emulator. Section 4 outlines the 4D–Var framework and the metrics used for the results evaluation. Results from twin experiments and with real observations are presented in Section ??, 5 and Section 6, respectively, followed by a discussion in Section 7 and conclusions in Section 8.

## 2 Physical model simulation and observations preprocessing of atmospheric forcings

In this section, we begin by introducing the geophysical sea-ice model, neXtSIM, which serves as the ground model for our results (See. ??). Following this, we describe the observations employed in the 4D–Var scheme. Specifically, we outline the observations simulated based on neXtSIM in See. 5.1, followed by real CS2SMOS retrievals in See. 6.1.

### 2.1 NeXtSIM sea-ice thickness

. We then present the atmospheric forcings used as additional inputs for the emulator. NeXtSIM is a state-of-the-art sea-ice model (Rampal et al., 2016) built around brittle rheologies (Girard et al., 2011; Dansereau et al., 2016). In the here-used simulations (Boutin et al., 2023), neXtSIM is employed with the brittle Bingham-Maxwell rheology (?) (Ólason et al., 2022).

to replicate the observed subgrid-scale behavior of sea ice. Originally run on a Lagrangian triangular mesh, neXtSIM's outputs are projected onto a Eulerian curvilinear grid, forming the basis of our surrogate model (Durand et al., 2024). Additionally, the sea-ice model is coupled with the ~~ocean component of the modeling framework NEMO~~Nucleus for European Modelling of the Ocean (NEMO) framework's ocean model, OPA (version 3.6, Madec et al., 1998; Rousset et al., 2015). For further information 95 about the model and its setup, we refer to Boutin et al. (2023).

In this study, we predict the sea-ice thickness with a neural network which is trained on simulations spanning from 2009 to 2016. The simulations are run on the regional CREG025 mesh configuration (Talandier and Lique, 2021), a regional subset of the global ORCA025 configuration developed by the Drakkar consortium (Bernard et al., 2006). The simulated area covers the Arctic and parts of the North Atlantic down to 27°N latitude, with a nominal horizontal resolution of 0.25° ( $\simeq 12\text{ km}$  in 100 the Arctic basin). The data is cropped in lower latitudes and areas in Eastern Europe and America where no sea ice is present, and then coarse-grained by averaging over a  $4 \times 4$  window, resulting in a final resolution of  $128 \times 128$  grid-cells. A simulated sea-ice thickness snapshot is displayed in Fig. 1a).

Let  $\mathbf{x} \in \mathbb{R}^{128 \times 128}$  be the sea-ice thickness. The normalized sea-ice thickness  $\tilde{\mathbf{x}}$  is then defined by

$$\tilde{\mathbf{x}} = \frac{\mathbf{x} - \mu_{\text{SIT}}}{\sigma_{\text{SIT}}}, \quad (1)$$

with  $\mu_{\text{SIT}}$  the globally averaged sea-ice thickness and  $\sigma_{\text{SIT}}$  the global standard deviation, computed over all the coarse-grained grid-cells in the training dataset (2009 to 2016). The subtraction and division are pointwise operations. Masking land-covered cells within the original  $128 \times 128$  grid-cells,  $N_z = 8871$  remain unmasked covered by either open water or sea ice. Hence, the data assimilation is performed on the 1-D state vector  $\tilde{\mathbf{x}}^{\text{1D}} \in \mathbb{R}^{N_z}$  which represents the normalized sea-ice thickness on the unmasked grid-cells. We will further drop the 1D superscript for the sake of readability.

We also consider the 2 m temperature (T2M), and the atmospheric  $u$ - and  $v$ -velocities in 10 m height (U10 and V10) from the ERA5 reanalysis dataset (Hersbach et al., 2020). Interpolated onto the native Eulerian curvilinear grid with nearest neighbors, forcings at time  $t$ ,  $t + 6\text{h}$  and  $t + 12\text{h}$  are then coarse-grained, normalized and added as predictors to the input of the neural network, as commonly done in sea-ice forecasting (Grigoryev et al., 2022). The normalization of all input fields in the neural network is a common practice to stabilize and speed up the training (Ioffe and Szegedy, 2015).

## 115 2.1 Observations

### 2.0.1 Simulated observations from neXtSIM

~~In the first approach, a twin experiment setup is employed, wherein synthetic observations are generated by adding noise to neXtSIM simulations from 2017 and 2018. Using the observation error variance  $\sigma_{\text{obs}}^2 = 0.4^2$ , we define several types of~~

~~perturbed observations,~~

$$\tilde{x}_{t,obs}^G = \tilde{x}_t + \epsilon^G, \quad \epsilon^G \sim \mathcal{N}(0, \sigma_{obs}^2),$$
$$\tilde{x}_{t,obs}^{LN} = \mu_{SIT} + \sigma_{SIT} x_t \exp(\epsilon^{LN}), \quad \epsilon^{LN} \sim \mathcal{N}\left(0, \sigma_{obs} - \frac{1}{2}\sigma_{obs}^2\right),$$
$$\tilde{x}_{t,obs}^{\text{cond-clipped}} = \min(\tilde{x}_t (1 + \epsilon^G), \min_{SIT}).$$

~~with  $\exp$  the exponential function. The Gaussian observation noise, as defined in Eq. (13a), is an idealized case, tailored to the common assumptions of 4D-Var, to test an adaptive inflation scheme, which will be defined later. Equation (13b) specifies a log-normal distribution for the noise, as more commonly encountered in sea-ice observations from satellites. Furthermore, a variant to log-normal noise is introduced in Eq. (13c) by adding a fraction of the sea-ice thickness and incorporating clipping to the normalized minimum ( $\min_{SIT}$ ). This approach ensures that the observations remain confined to the physical bound of sea ice, unlike Gaussian noise. Examples of the different noises are shown in Fig. 1b)–d).~~

~~Snapshots of neXtSIM SIT (a) and different type of observations (b) Gaussian noise, (c) log-normal noise (LN) and (d) conditioned noise (cond-clipped). The colorbar for panels (b), (c) and (d) is shared and displayed on the right.~~

~~These noise definitions yield different noise magnitudes. The log-normal noise, defined in Eq. (13b), provides a more significant spread, especially for thicker ice. In average, the log-normal noise definition results in a standard deviation of 0.35 m, because of the skewness of the log-normal law, whereas the conditioned noise, defined in Eq. (13c), results in a smaller standard deviation of 0.29 m.~~

**2.0.1 Real observations: combined CryoSat2-SMOS retrieval**

~~The dataset of CS2SMOS (Ricker et al., 2017) retrievals provides real observations that are assimilated into our surrogate model. The retrievals merge observations from CryoSat-2 (Kurtz and Harbeck, 2017), known for its observations of thick and perennial sea ice, and from SMOS (Tian-Kunze et al., 2014), used to infer the thickness of thin ice. Merged weekly to account for the different temporal resolution of CryoSat-2 and SMOS observations, the retrievals are available in a daily moving window average. Note that the CS2SMOS is the result of Kriging and has been considerably smoothed in the process, even when compared to a weekly average of neXtSIM, as illustrated in Fig. 6.~~

~~CS2SMOS retrievals are only available on grid-cells covered by sea ice, and no information is available on grid-cells with open water. This creates a temporally changing mask, and we assume that grid-cells without information contain no sea ice.~~

~~Additionally, the CS2SMOS retrievals come with their own errors and uncertainties (Ricker et al., 2017). Based on the diagnostics of Desroziers et al. (2005), Xie et al. (2018) proposed an empirical formula for the observation error variance  $\sigma_{obs,CS2SMOS}^2$ , as an increasing function of ice thickness  $h_{ice}$ ,~~

$$\sigma_{obs,CS2SMOS}^2 = \begin{cases} \min(0.2, 0.02e^{1.8(h_{ice}-3)}) & \text{if } h_{ice} > 3\text{m}, \\ \max(0.02, 0.1e^{-1.5h_{ice}}) & \text{otherwise.} \end{cases}$$

This observation error variance is also used in Cheng et al. (2023b). We will rely on this assessment to introduce observation error statistics for the real observation setup. Note that, unlike the usual approach in data assimilation where the model state is projected onto the observation space using  $\mathcal{H}$ , we simplify the process by doing the other way around. In a preprocessing step, real observations are interpolated onto the model space. This is feasible because the observations are at a higher resolution yet smoother than the forecasts with our surrogate model.

Difference (right) between CS2SMOS (left) and neXtSIM SIT (middle). CS2SMOS is interpolated on neXtSIM reduced grid. neXtSIM SIT is averaged over one week in order to mimic CS2SMOS weekly averaging.

**3 Surrogate model**

In this section, we describe our surrogate model, which has the same structure as the emulator previously developed in Durand et al. (2024), with the only update being that it is trained to account for the positivity of sea-ice thickness.

The surrogate model  $g_\theta$  predicts the full sea-ice thickness  $\tilde{\mathbf{x}}_{t+\Delta t}$  with a  $\Delta t = 12$  h lead time. The neural network  $f_\theta$  with its weights and biases  $\theta$  is trained to predict the evolution of the SIT after  $\Delta t = 12$  h based on the initial conditions  $\tilde{\mathbf{x}}_t$  and given atmospheric forcings  $\mathbf{F}_t$ .  $f_\theta(\tilde{\mathbf{x}}_t, \mathbf{F}_t) \approx \tilde{\mathbf{x}}_{t+\Delta t} - \tilde{\mathbf{x}}_t$ . Added to the initial conditions  $\tilde{\mathbf{x}}_t$ , this results in the prediction of the full SIT field  $\tilde{\mathbf{x}}_{t+\Delta t}$ .

$$\tilde{\mathbf{x}}_{t+\Delta t} = g_\theta(\tilde{\mathbf{x}}_t, \mathbf{F}_t) \quad (2a)$$

$$= \text{Relu}(\tilde{\mathbf{x}}_t + f_\theta(\tilde{\mathbf{x}}_t, \mathbf{F}_t)), \quad (2b)$$

with the point-wise activation function,  $\text{Relu}(\tilde{\mathbf{x}}) = \max(\min_{\text{SIT}} \tilde{\mathbf{x}}, \tilde{\mathbf{x}}) \text{Relu}(\tilde{\mathbf{x}}) = \max(\text{SIT}_{\text{min}}, \tilde{\mathbf{x}})$ , limiting the output to the lower physical bound in the normalized space. Note that  $f_\theta$  is an intermediate emulator. As similarly done in Durand et al. (2024), the idea is to simplify the problem by focusing on the tendencies that happen in 12 h instead of predicting directly the full state, where small changes can be harder to predict. By splitting the whole learning process, we ensure that the emulator learns the SIT evolution, and in a second time that  $g_\theta$  respects the positiveness constraint.

The neural network is trained with a mean-squared error loss between the predicted sea-ice thickness and the targeted sea-ice thickness as simulated by neXtSIM. The main part of the loss function is defined by a pixel-wise mean-squared error (MSE) on all  $N_x \times N_y$   $N_x \times N_y = 128 \times 128$  grid-cells, multiplied by the land-sea mask,

$$\mathcal{L}_{\text{local}}(\mathbf{x}, \hat{\mathbf{x}}) = \text{MSE}(\mathbf{x}, \hat{\mathbf{x}}) = \frac{1}{N_x \cdot N_y} \sum_i^{N_x} \sum_j^{N_y} (x_{i,j} - \hat{x}_{i,j})^2. \quad (3)$$

Note that, to simplify the equation, we did not include here the land-land-sea mask, which is applied in the numerical implementation to compute the loss solely on grid cells with open ocean and sea ice. To address the systematic bias of the surrogate model and to mitigate its influence, which is already accounted for in the MSE loss, we introduce an additional penalty term

to the loss function,

$$\mathcal{L}_{\text{global}}(\mathbf{x}, \hat{\mathbf{x}}) = \left( \frac{1}{N_x \cdot N_y} \sum_i^{N_x} \sum_j^{N_y} (x_{i,j} - \hat{x}_{i,j}) \right)^2. \quad (4)$$

Note in particular that the loss in Eq. (4) is squared after averaging, differing from Eq. (3). The total loss is then given by

$$\mathcal{L}(\mathbf{x}, \hat{\mathbf{x}}) = \mathcal{L}_{\text{local}}(\mathbf{x}, \hat{\mathbf{x}}) + \lambda \mathcal{L}_{\text{global}}(\mathbf{x}, \hat{\mathbf{x}}), \quad (5)$$

with  $\lambda$  weighting the two terms. First, we pre-train the model  $f_\theta$ , omitting the clipping in Eq. (2b) during training, with a first loss. As in Durand et al. (2024), we use  $\lambda = 100$ . Trained until convergence, we select the best model in the validation dataset. Secondly, we fine-tune  $g_\theta$  to account for the clipping with a second loss. Here, we select  $\lambda = 10$  as penalty weight, striking a good balance between  $\mathcal{L}_{\text{local}}$  and  $\mathcal{L}_{\text{global}}$  which have a different magnitude during second training. Results of the training and inference of the surrogate are presented in Appendix A.

**4 Four-dimensional variational data assimilation**

In this section, we describe the experimental setup of our 4D–Var, which is based on the surrogate model and its adjoint, as previously discussed in Sec. 3. Note that we are focusing on full 4D–Var rather than incremental 4D–Var.

#### 4.1 4D–Var setup

The ~~length of the DAW is set to  $N_{\text{daw}} = 16$  days which corresponds to 32 iterations of the surrogate model. In the twin experiment setup, observations are acquired every 2 days (every  $N_t = 4$  iterations). The truth is given by neXtSIM, and the forecast model is our data-driven emulator of neXtSIM.  $\tilde{y}_k$  represents the  $k$ -th observation observation at time  $k$ ,  $\tilde{x}^b$  represents the background state,  $\tilde{x}^a$  represent the analysis and  $\tilde{x}_0$  the first guess.  $\mathbf{B}$  is the background error covariance matrix and its specific parameterizations will be described below.  $\mathbf{R}$  is the observation error covariance matrix  $\mathbf{R}$  and is defined by  $\mathbf{R} = \sigma_{\text{obs}}^2 \mathbf{I}$ , with  $\mathbf{I}$  the identity matrix. To initialize the cycling of the data assimilation, we start with a field as simulated by neXtSIM for the 1st of January 2016, which is contained in our training dataset and which can be seen as sample from the climatology for the starting day.~~

For twin experiments, observations are generated directly in the model space, requiring no special preprocessing, as opposed to real observations, as explained in Sec. 6.1. ~~After that, in both cases, the~~ The observation operator  $\mathcal{H}$  is simply defined as a diagonal matrix  $\mathbf{H}$ .

In this study, the 4D–Var is cycled across  $N_{\text{cycle}}$  cycles. The state at the end of each DAW is used as first guess and background state for the next DA cycle. To evaluate the 4D–Var system, forecasts are run for 45 days after the end of the DAW in the twin experiment case, and for 9 days in the real observations case.

Two types of 4D–Var are evaluated. 4D–Var-diag, which ~~correspond~~ corresponds to the use of a diagonal matrix as ~~background matrix covariance matrix of background errors~~  $\mathbf{B}$  (see Sec. 4.2), and 4D–Var-EOF in which the minimization is carried out in 205 the empirical orthogonal function (EOF) space (see Sec. 4.3).

## 4.2 4D–Var with diagonal $\mathbf{B}$ matrix: 4D–Var-diag

The cost function associated to the 4D–Var minimization problem is

$$\mathcal{J}(\tilde{\mathbf{x}}_0) = \mathcal{J}^b + \mathcal{J}^o \quad (6a)$$

$$= \frac{1}{2} \left\| \tilde{\mathbf{x}}_0 - \tilde{\mathbf{x}}^b \right\|_{(\lambda_{\inf}^2 \mathbf{B})^{-1}}^2 + \frac{1}{2} \sum_{k=1}^K \left\| \tilde{\mathbf{y}}_k - \mathbf{H}_k \tilde{\mathbf{x}}_k \right\|_{\mathbf{R}_k^{-1}}^2, \quad (6b)$$

with

$$\tilde{\mathbf{x}}_k = \mathcal{M}_{0 \rightarrow k \cdot \Delta t}(\tilde{\mathbf{x}}_0) = \underbrace{g_\theta \circ \cdots \circ g_\theta}_{k\text{-times}}(\tilde{\mathbf{x}}_0), \quad (7)$$

and where  $\|\mathbf{x}\|_{\mathbf{A}} = \sqrt{\mathbf{x}^\top \mathbf{A} \mathbf{x}}$  is the Mahalanobis norm, and  $\mathbf{B}$  is defined by  $\mathbf{B} = \sigma_b^2 \mathbf{I}$ . Through all experiments,  $\sigma_b^2 = 0.4^2$  (non-dimensional, as  $\tilde{\mathbf{x}}$  has been normalized). The coefficient  $\lambda_{\inf}$  is a background multiplicative inflation term for background errors, which is set to 1 when no inflation is used and is further described in Appendix E. Note that the observation error covariance matrices  $\mathbf{R}_k$  are not inflated because, with diagonal matrices, this is equivalent to background inflation, as far as the cost function minimization is concerned. Within the DAW of 16 days, with observations taken every second day starting on day 2, the total number of observations observation times is  $K = 8$ . The results of the minimization of the cost function is the analysis  $\mathbf{x}^a$ .

## 4.3 4D–Var projected onto the EOFs basis: 4D–Var-EOF

Empirical orthogonal functions (EOFs) are a set of orthogonal state vectors derived from data, which form a basis of the full state space. Details about the computation and the analysis of the EOFs are given in Appendix B. A reduced strategy to enhance 4D–Var by projecting onto the EOFs has been proposed by Robert et al. (2005) for ocean models. The projection onto the EOFs enables access to cross-covariances and improves may improve the numerical conditioning of the B-matrix, thereby enhancing the minimization of the cost function minimization of  $\mathcal{J}$ . The minimization is carried out with respect to the control variable  
$\mathbf{w} \in \mathbb{R}^m$  defined by the projection of the vector  $\tilde{\mathbf{x}}$  onto the matrix  $\varphi_m \in \mathbb{R}^{N_z \times m}$ :

$$\tilde{\mathbf{x}} = \bar{\mathbf{x}} + \varphi_m \mathbf{w}, \quad (8)$$

with  $\bar{\mathbf{x}}$  representing the temporal average sea-ice thickness field over the dataset used to construct the EOFs and  $m$  standing for the truncation index corresponding to the number of EOFs which are kept in the definition of the minimization subspace. The goal is to run the 4D–Var minimization in this affine truncated space spanned by the  $\varphi_m$ .

In this subspace, the cost function reads, at cycle  $n$ ,

$$\mathcal{J}(\mathbf{w}_0) = \frac{1}{2\lambda_{\inf}^2} \left\| \mathbf{w}_0 - \mathbf{w}_0^b \right\|^2 + \frac{1}{2} \sum_{k=1}^K \left\| \tilde{\mathbf{y}}_k - \mathbf{H}_k \tilde{\mathbf{x}}_k \underbrace{g_\theta^{k \times N_f}}_{\text{choiced value}}(\bar{\mathbf{x}} + \varphi_m \mathbf{w}_0, \mathbf{F}_{k \times N_f \mapsto 0}) \right\|_{\mathbf{R}_k^{-1}}^2. \quad (9)$$

The details of the 4D–Var cost function computation are presented in Alg. C1 and Alg. C2. The result of this minimization is the control variable at the beginning of the DAW,  $\mathbf{w}_0^a$ . The choice value of the truncation index is set to  $m = 7000$  for all experiments and is further discussed in the Appendix B while further details on the optimization are given in Appendix C.

In this section, the 4D-Var results obtained first on simulated observations (Sec. 5), then on real observations (Sec. 6) are presented.

#### 4.1 Metrics for the evaluation of the experiments

To evaluate the efficiency of the data assimilation scheme, we compute the root-mean-squared error (RMSE) between the 240 predicted state performed by applying the emulator with the analysis as initial state,  $\mathbf{x}^f(t) = \mathcal{M}_{0 \rightarrow t\Delta t}(\mathbf{x}_0^a)$ , and the truth  $\mathbf{x}_t$ , which corresponds to the neXtSIM simulation in the twin experiment case, and to CS2SMOS fields in the real observations case, for each cycle  $n$ , at the lead time  $t$ , and over all unmasked pixels  $i \in N_z$ . Inside the DAW (up to 16 days for the twin experiments case and 8 days for the real observations case), this corresponds to an analysis RMSE and afterwards a forecast RMSE.

$$\text{RMSE}_n(t) = \sqrt{\frac{1}{N_z} \sum_{i=1}^{N_z} (\mathbf{x}_{i,n}^f(t) - \mathbf{x}_{i,n}^t(t))^2}. \quad (10)$$

This RMSE is defined for each cycle of each experiment, and can then be averaged across all cycles to get the mRMSE, which becomes a function depending on the lead time  $t$  only.

$$\text{mRMSE}(t) = \frac{1}{N_{\text{cycle}}} \sum_{n=1}^{N_{\text{cycle}}} \text{RMSE}_n(t). \quad (11)$$

For the evaluation of CS2SMOS assimilation, see Sec. 6, we use two additional metrics, the bias error,

$$\text{bias}_n(t) = \frac{1}{N_z} \sum_{i=1}^{N_z} \mathbf{x}_{i,n}^f(t) - \frac{1}{N_z} \sum_{i=1}^{N_z} \mathbf{x}_{i,n}^t(t), \quad (12)$$

and the Ice Integrated Edge Error (HEE-IIE<sub>SIT</sub>), as initially introduced by Goessling et al. (2016), but slightly modified by using sea-ice thickness instead of sea-ice concentration as the threshold. Specifically, we define a metric that counts the grid cells where the surrogate model disagrees with CS2SMOS on the presence of sea ice. A grid cell is considered to be covered by sea ice if the thickness exceeds 0.1 m (Durand et al., 2024) (cf Durand et al., 2024, B1 for the threshold justification), analogous to the sea-ice concentration threshold of 0.15 defined in Goessling et al. (2016). The HEE-IIE<sub>SIT</sub> is expressed as a 255 fraction of grid cells (not multiplied by the sea-ice covered area).

#### 4.2 Twin experiments

the total area (km<sup>2</sup>) of disagreement between the observations and the forecast.

### 5 Twin experiments

In this section, the results from the twin experiments are discussed, with the observations simulated as explained presented in Sec. 5.2, after the description of the simulated observations in Sec. 5.1. In-

## 5.1 Simulated observations from neXtSIM

In the first approach, a twin experiment setup is employed, wherein synthetic observations are generated by adding noise to neXtSIM simulations from 2017 and 2018. Using the observation error variance  $\sigma_{\text{obs}}^2 = 0.4^2$  (non-dimensional), we define several variants of perturbed observations.

$$\tilde{x}_{t,\text{obs}}^G = \tilde{x}_t + \epsilon^G, \quad \epsilon^G \sim \mathcal{N}(0, \sigma_{\text{obs}}^2), \quad (13a)$$

$$\tilde{x}_{t,\text{obs}}^{\text{LN}} = \frac{x_t \exp(\epsilon^{\text{LN}}) - \mu_{\text{SIT}}}{\sigma_{\text{SIT}}}, \quad \epsilon^{\text{LN}} \sim \mathcal{N}\left(0, \sigma_{\text{obs}}^2 - \frac{1}{2}\sigma_{\text{obs}}^2\right), \quad (13b)$$

$$\tilde{x}_{t,\text{obs}}^{\text{cond-clipped}} = \min(\tilde{x}_t(1 + \epsilon^G), \text{SIT}_{\text{min}}). \quad (13c)$$

with  $\exp$  the exponential function. The Gaussian observation noise as defined in Eq. (13a), is an idealized case tailored to the common assumptions of 4D–Var, to test an adaptive inflation scheme which will be defined later. Equation (13b) specifies a log-normal distribution for the noise, as more commonly encountered in sea-ice observations from satellites (Landy et al., 2020). Furthermore, a variant to log-normal noise is introduced in Eq. (13c) by adding a fraction of the sea-ice thickness and incorporating clipping, based on  $\text{SIT}_{\text{min}}$  the corresponding 0m thickness in the normalized space. This approach ensures that the observations remain confined within the physical bounds of sea ice, unlike Gaussian noise, but similarly to log-normal noise. Examples of the different noises are shown in Fig. 1b)–d).

These noise definitions yield different noise magnitudes. The log-normal noise, defined in Eq. (13b), provides a more significant spread, especially for thicker ice. In average, the log-normal noise definition results in a standard deviation of 0.35 m, because of the skewness of the log-normal law, whereas the conditioned noise, defined in Eq. (13c), results in a smaller standard deviation of 0.29 m.

## 280 5.2 Results

The length of the DAW is set to  $N_{\text{daw}} = 16$  days which corresponds to 32 iterations of the surrogate model. In the twin experiment setup, observations are acquired every 2 days (every  $N_f = 4$  iterations). In all twin experiments, we initialize the data assimilation cycles with the same past field, first data assimilation cycle with a past neXtSIM SIT field from the 1st of January 2016. In this section, the multiplicative inflation coefficient is set to 1. Experiments are conducted throughout years 2017 and 2018, which are after the training dataset (years 2009 – 2016). To compute averaged results, only the year 2018 is used, i.e. 2017 is used as spin-up. The 4D–Var is run on a single trajectory starting the January 1st 2017, for  $45 N_{\text{cycle}} = 45$  cycles, until December 22th 21th, 2018.

As shown in Tab. 1, the choice of noise distribution—whether Gaussian, log-normal, or cond-clipped as defined in Sec. 5.1, influences the efficiency of the assimilation process. However, all experiments remained stable, meaning that there was no divergence in the results. While the assimilation of perfect observations yields the best results, the different noise definitions produce comparable outcomes. Using the cond-clipped perturbations provide the best RMSE among the three different type types of perturbations. This could be linked to the lower RMSE of its observations. As shown in Tab. 1, projecting the 4D–Var

**Figure 1.** Snapshots of neXtSIM SIT (a) and different type of observations (b) Gaussian noise, (c) log-normal noise (LN) and (d) conditioned noise (cond-clipped). The colorbar for panels (b), (c) and (d) is shared and displayed on the right.

**Table 1.** Comparison of RMSE for different types of simulated observation noise (cf. Sec. 5.1) for the two types of 4D–Var algorithms (with diagonal  $\mathbf{B}$  matrix and with projection onto the EOFs). Results are presented with the mRMSE computed inside the DAW across 2018. The RMSEs between neXtSIM and the perturbed observations, are outlined in the second column. They correspond to the averaged RMSE between the observations and their associated non-perturbed SIT field.

| mRMSE (m)          | Observations | 4D–Var-diag | 4D–Var-EOF |
|--------------------|--------------|-------------|------------|
| No noise           | 0.000        | 0.305       | 0.256      |
| Gaussian noise     | 0.638        | 0.321       | 0.272      |
| LN noise           | 0.587        | 0.333       | 0.281      |
| Cond-clipped noise | 0.527        | 0.318       | 0.264      |

onto the EOFs yields improvements in all cases, with relative improvements in the range 15% – 17% for the different types of noise. This improvement can be attributed to the preconditioning of non-diagonal terms in the  $\mathbf{B}$  matrix and its non-diagonal terms. This systematic improvement is also observed in Fig. 2 during forecast. When extending the forecast beyond the DAW, the advantage of the 4D–Var-EOF over the 4D–Var-diag remains noticeable but diminishes as the lead time increases. On average, both forecasts show an improvement of 0.8 m over the emulator’s free run (initialized on January 1st, 2017) across all lead times. Additionally, when comparing the 4D–Var-EOF forecast to the emulator forecast (initialized with perfect conditions at the end of each DAW, red dashed curve), we observe a slight improvement of 1.1 cm, demonstrating a gain in forecast skill with our assimilation system comparable RMSE, showing the stability of the forecast produced by the analysis.

Figure 3 displays fields to illustrate the benefit of data assimilation over a cycle. The first guess field is smoother because of the emulator, whereas the analysis is actually noisier than the truth, because of the observations’ noise. The largest corrections are applied to the MIZ to correct its position, especially in the Beaufort Sea, Chukchi Sea, and Hudson Bay. In the other places, we observe a positive correction, which is consistent with the negative bias ( $\simeq -0.015$  m) of the surrogate model at the time of the depicted cycle.

As seen in Fig. 4, at the start of each cycle, the initial analysis RMSE is generally lower than the first guess RMSE, which corresponds to the end of the previous DAW, indicating that the analysis improves over the first guess, of 2 cm in average across all cycles. In the specific case on late February 2018 (cycle 26), large differences between the truth and the analysis are located near the Canadian Archipelago, leading to a higher RMSE compared to the first-guess RMSE, where such differences do not occur. We hypothesize that, in order to minimize the RMSE throughout the DAW, the 4D–Var removes sea ice in this region. This is consistent with the emulator’s positive bias during this period. The analysis from Fig. 4(top) reveals a strong seasonality in results, with RMSE peaking in May, dropping, then rising again in December. These peaks align July. The peak aligns with significant changes in sea-ice extent: decreasing in summer (May) and increasing rapidly by winter (December) at this period, with an important decrease due to the warmer temperatures in the Arctic. This suggests that 4D–Var possibly struggles more with dynamic shifts. See additional multi-year results in Appendix. D. Additionally, the bias error of the emulator (Fig. 4,

**Figure 2.** Cycle-averaged mRMSE of  $\tilde{x}_f$  over 2018 during forecast stage. The dashed red line represents the free run of the emulator started at the end of each DAW (with perfect initial conditions, PIC). The mRMSE of the 4D-Var-EOF forecast is shown in purple, and the 4D-Var-diag forecast is shown using in grey. Cond-clipped noise is used in both cases. The black dotted line corresponds to the emulator free run, initialized on January 1st, 2017.

lower panel), indicative of model error, shows a similar seasonal pattern, transitioning from a positive to negative bias around summer and reverting in December. The key factor affecting assimilation accuracy appears to be not just the amplitude but also the seasonal variation of this bias.

### 5.3 CS2SMOS assimilation

An important factor influencing the quality of the assimilation is the frequency of the observations. We denote the number of observations during one cycle of 16 d (32 model iterations) as  $N_{\text{obs}}$ . As shown in Fig 5, when there are too few observation times per cycle, no improvement is observed in the analysis compared to the forecast at the end of the previous cycle (this is the case for both 2 and 4 observation times per cycle). Once 8 observation times per cycle are reached, this divergence disappears, which is why we choose 8 observation times per cycle in our setup.

6 CS2SMOS assimilation

In this section, the results from the real observations are presented in Sec. 6.2, after the description of the observations used in Sec. 6.1.

#### 6.1 Real observations: combined Cryosat2-SMOS retrieval

### 4D–Var, 25th cycle

**Figure 3.** Fields of the SIT in the 10th beginning of the 25th cycle of the DA, corresponding to 2017-06-10 2018-02-05, are shown in the 4D-Var-EOF case. The upper left panel represents the first guess, which is the output of the forecast from the previous minimization. The upper right panel corresponds to the analysis of the 10th-25th cycle. For comparison, the associated neXtSIM field, considered as the truth, is displayed in the lower left panel. Note that these three fields share the same colormap and scale. The lower right panel shows the analysis increment, which represents the analysis minus first guess.

**Figure 4.** Time evolution of the 4D-Var-EOF analysis RMSE (inside the DAW) in 2018 (upper panel), with cond-clipped simulated noise. Corresponding bias error of the emulator is represented below, as defined in Eq. (A2) with a 16 days lead time, with a new forecast starting every 6 h, at the given time of the  $x$ -axis.

**Figure 5.** 2018 RMSE of 4D-Var-EOF depending on the number of observations in every assimilation cycle in the twin experiment case. Curves indicated the RMSE of the 4D-Var analysis with regard to neXtSIM SIT, with 2 observations per cycle (purple curve), 4 observations per cycle (blue curve), 8 observations per cycle (gray curve) and 16 observations per cycle (green curve). The dashed grey line correspond to the averaged RMSE between log-normal (LN) noise and the truth.

**Figure 6.** Difference (right) between daily CS2SMOS (left) and neXtSIM SIT (middle). CS2SMOS is interpolated on neXtSIM reduced grid. neXtSIM SIT is averaged over one week in order to mimic CS2SMOS weekly averaging, centered on 2018-01-04.

The dataset of CS2SMOS (Ricker et al., 2017) retrievals provides real observations. The retrievals merge observations from 330 CryoSat-2 (Kurtz and Harbeck, 2017), known for its accurate observations of thick and perennial sea ice, and from SMOS (Tian-Kunze et al., 2014), used to infer the thickness of thin ice. Merged weekly to account for the different temporal resolution of CryoSat-2 and SMOS observations, the retrievals are available as daily moving window average. Note that the CS2SMOS is the result of Kriging and has been considerably smoothed in the process, even when compared to a weekly average of neXtSIM, as illustrated in Fig. 6.

CS2SMOS retrievals are only available on grid-cells covered by sea ice, and no information is available on grid-cells with open water. This creates a temporally changing mask, and we assume that grid-cells without information contain no sea ice.

Additionally, the CS2SMOS retrievals come with their own errors and uncertainties (Ricker et al., 2017). Note that Nab et al. (2025) showed that modifying SIT observation uncertainties introduces significant sensitivities during SIT assimilation. Based on the diagnostics of Desroziers et al. (2005), Xie et al. (2018) proposed an empirical formula for the observation error variance  $\sigma_{\text{obs,CS2SMOS}}^2$  as an increasing function of ice thickness  $h_{\text{ice}}$ , with the coefficient 0.2, 0.02, 0.1 in  $\text{m}^2$ , 3 in  $\text{m}$  and 1.5 without unit.

$$\sigma_{\text{obs,CS2SMOS}}^2 = \begin{cases} \min(0.2, 0.02e^{1.8(h_{\text{ice}}-3)}) & \text{if } h_{\text{ice}} > 3\text{m}, \\ \max(0.02, 0.1e^{-1.5h_{\text{ice}}}) & \text{otherwise.} \end{cases} \quad (14)$$

This observation error variance is also used in Cheng et al. (2023a). We will rely on this assessment to introduce observation error statistics for the real observation setup. Note that, unlike the usual approach in data assimilation where the model state is projected onto the observation space using  $\mathcal{H}$ , we simplify the process by doing the other way around. In a preprocessing step, real observations are interpolated onto the model space, making them retrievals. This is feasible because the observations are at a higher resolution in their native grid, and then coarse-grained by a factor 2. Even in their original resolution, they remain smoother than the forecasts of our surrogate model.

## 6.2 Results

In this section, instead of assimilating simulated observations, we assimilate CS2SMOS retrievals daily (every two iterations of the emulator) within an 8-day window for  $N_{cycle} = 20$  cycles. We use the observations for the winter 2020-2021, since there are no CS2SMOS retrievals during summer. We consider the truth now as CS2SMOS. Note that ground truth is a quantity that is assimilated, and hence that the analysis score should not be over-interpreted. To initialize the cycling of the data assimilation, we start with a field as simulated by neXtSIM for the 10th of October 2016, the CS2SMOS retrieval from the date of the first assimilation cycle. The observation error variance used is defined in Eq. (14). Only results performed with the 4D-Var-EOF are presented in this section. Note that no multiplicative inflation scheme is used.

In order to compare the results to a practical forecast benchmark, we use the past forecasts from neXtSIM-F (Williams et al., 2021), a forecasting system that consists of a stand-alone version of neXtSIM, forced by the TOPAZ ocean forecast (Sakov et al., 2012) and ECMWF atmospheric forecasts. The past forecasts have been obtained in 2023, corresponding to the 360 version of the forecasting system released in November 2023 (European Union-Copernicus Marine Service, 2020). **NeXtSIM-F assimilates** In neXtSIM-F, the model is nudged to CS2SMOS sea-ice thickness observations weekly with a simple nudging. It produces a 9-day forecast, which we will use for systematically use for numerical comparison with our data assimilation scheme, starting the neXtSIM-F forecast at the end of each DAW. Both data assimilation systems are compared directly to CS2SMOS observations for forecasts beyond the DAW.

The 4D-Var analysis is using atmospheric reanalysis  $\mathbf{F}$  from ERA5 as input for the emulator. For the sake of fairness, for the 9-day forecast that are run beyond the DAW, we use atmospheric forcings from the ECMWF atmospheric model HRES, which provides 10-day forecasts at a 16 km resolution. **By contrast, ERA5 is a reanalysis product that assimilates observations, making it unsuitable for an operational forecasting setup where the aim is to predict the future. Therefore, following the approach used in neXtSIM-F, we rely on atmospheric forecasts as forcings of the emulator during the forecast period.** These 370 forecasts are interpolated onto the neXtSIM grid and normalized using the same processing method as the ERA5 forcings preparation, replacing them when applying the emulator during forecast.

As seen in Fig. 7, the assimilation of real data into our 4D-Var works and yields RMSEs similar to those of the neXtSIM-F assimilation forecast. The free run, initialized with a SIT field from October 2018 (due to the lack of simulation outputs in 2020), produces a stable trajectory but significantly deviates from the CS2SMOS fields. Assimilating real observation yields a substantial decrease in RMSE, of  $-0.49$  m during forecast. Results after the 9-day forecast are compared with the neXtSIM-F 9-day forecast and show similar outcomes. **In-On average, neXtSIM-F has a 0.34 m RMSE while our forecast has a 0.36 m RMSE.** Initially, neXtSIM-F exhibits lower RMSEs; however, we see improved results in term of forecast towards the last cycles of the assimilation. The RMSE score metric penalizes the high level of details of the neXtSIM model more than the emulator that smooths gradually with time, an effect known as "double penalty" in weather forecasting.

As seen in the twin experiment results, especially with the Fig. 4, we can infer that our data assimilation system is less efficient during periods of strong dynamic change. This is also shown here, with better results after January, at the end of the refreezing period when comparing the end of each cycle forecast (end of the purple dashed lines) with corresponding neXtSIM-

**Figure 7.** RMSE results for CS2SMOS assimilation across several DAWs throughout the full CS2SMOS observation period in 2020-2021 are shown. The black dotted line represents the free run of the emulator through all cycles, initialized with a SIT field from October 2018. The solid purple line corresponds to the analysis of the 4D–Var over the DAW, while the associated dashed purple lines represent the additional forecasts using ECMWF atmospheric forecasts for 9 days. The RMSE values from neXtSIM-F, corresponding to a 9-day forecast, are displayed as blue dots and should be compared with the end of each corresponding dashed line. All RMSE values are computed with CS2SMOS considered as the truth.

F. Conversely, when the system is less dynamic after the refreezing period, it faces fewer difficulties in predicting the optimal state. Yet, the high initial RMSEs could also be linked to the spin-up of the assimilation. Similarly, to evaluate our assimilation framework, we use the bias error and the HEE–IIE<sub>SIT</sub> as defined in Sec. 4.1 in the same data assimilation cycles, see Fig. 8. The bias error of the trajectory is significantly reducedabsolute value of the bias of the emulator free run is generally reduced, with the exception of December 2020, where the free run bias is smaller than the forecast bias. At the end of the forecast, the assimilation run brings a bias reduction of 6.7 cm compared to the free run. The average bias of the assimilation run compared to CS2SMOS at the end of the forecast is  $-0.22$  cm, while the bias of neXtSIM-F is  $-1.5$  cm. During the forecast, the bias systematically decreases, which might indicate that the emulator considers the analysis to increaseThe biases at the beginning of the DAWs are systematically positive and then generally decrease. Since the emulator tends to reduce the amount of sea iceneXtSIM, as indicated by its negative bias, the analysis overestimates sea ice to compensate for this loss. NeXtSIM, on which the emulator is trained, has a different SIT distribution than CS2SMOS, which could explain this phenomenon. The HEE, as it can be seen with the different order of magnitude of the bias between the emulator initialized with neXtSIM fields in Fig. 4b) (around 0.02 m) and the emulator initialized with CS2SMOS (around 0.1 m). The IIE<sub>SIT</sub> serves as a reliable indicator of how accurately the MIZ is positioned. The IIEE of neXtSIM-F is 5.29% and is slightly better than that of the assimilation run (5.88%). At the end of each forecast, the assimilation run shows a 15% improvement in IIEE compared to the free run, highlighting a significant enhancement in MIZ positioning achieved through the 4D–Var-EOF assimilation.

The data assimilation analysis acts as a bias correction for the emulator. However, it relies on smooth observations, thereby losing the small-scale information available in neXtSIM and neXtSIM-F, as illustrated in Fig. 9. In Fig. 3, we observe that the first guess, corresponding to the end of the previous cycle forecast, appears smoother than the truth. This smoothing is linked to the deterministic nature of the emulator (Durand et al., 2024), as it optimizes its MSE loss by smoothing fine-scale

**Figure 8.** Bias (upper panel) and  $\text{IIEE}_{\text{SIT}}$  (lower panel) results for CS2SMOS assimilation across several DAWs throughout the full CS2SMOS observation period in 2020–2021 are shown. The black dotted-dashed line represents the free run of the emulator through all cycles, initialized with a SIT field from October 2018. The solid purple line corresponds to the analysis of the 4D–Var over the DAW, while the associated dashed-dotted purple lines represent the additional forecasts using ECMWF atmospheric forecasts for 9 days. The bias errors and IIEE from neXtSIM-F, corresponding to a 9-day forecast, are displayed as blue dots and should be compared with the end of each corresponding dashed line. All bias errors and IIEE are computed with CS2SMOS considered as the truth.

dynamics. While in the twin experiments, the use of observations closely aligned with neXtSIM could help recover some small-scale dynamics, smooth observations result in a complete loss of these finer details. Nevertheless, our data assimilation 405 system effectively acts as a model error correction mechanism.

## 7 Discussion

We have demonstrated that using an emulator as forecast model in a 4D–Var framework for sea-ice forecasting is feasible, thanks to its numerical efficiency and auto-differentiability. However, this approach raises several important questions: one key observation is that while replacing the physics-based model with an emulator allows for the benefits of adjoint optimization 410 in a 4D–Var system, the success of the data assimilation is inherently tied to the emulator’s accuracy. As illustrated in Fig. 3, the emulator tends to smooth out SIT during forecasts extending up to 16 days, a behavior previously noted in Durand et al. (2024).

In the case of real observations, we observe a significant bias correction in the assimilation run. Yet, it is worth noting that this study does not consider weak-constraint 4D–Var, which incorporates model error into the cost function minimization.

**Figure 9.** Visualization of the forecast for 2021-03-26 is shown. The upper left panel displays the neXtSIM-F 9-day forecast, while the upper right panel shows the 9-day forecast from the 4D–Var. The lower left panel presents the CS2SMOS observations, and the lower right panel illustrates the difference between the 4D–Var forecast and the CS2SMOS observations.

However, we ~~can infer speculate~~ that improving the emulator’s quality—addressing both bias and smoothing issues—would result in more accurate 4D–Var analyses.

An analysis comparing the numerical efficiency of the 4D–Var method between 4D–Var-diag and 4D–Var-EOF is presented. For twin experiments, all computations were executed on a single NVIDIA A100 SXM4 80 GB GPU. On average, over a full data assimilation run of 45 cycles, 4D–Var-EOF takes 155 s per cycle, while 4D–Var-diag takes approximately 229 s per 420 cycle, making the 4D–Var-EOF around 32% faster. Although the gradient computation times differ slightly between the two methods — 1.66 s for 4D–Var-EOF versus 1.36 s for 4D–Var-diag — the forward pass through the DAW is similar, with times of 578 ms and 552 ms, respectively. The computation of the gradient involves the emulator adjoint evaluation and reshaping the one-dimensional vector of assimilated grid cells into the two-dimensional field required to run the emulator, as well as computing the cost function terms gradients. Note that the emulator adjoint is evaluated 8 times in each DAW. Interestingly,

while individual operations are faster in 4D–Var-diag (as expected due to the absence of projections to and from the EOF basis), the complete cycles are faster with the EOF-based method. In other words, fewer iterations of the L-BFGS optimizer are required on average for 4D–Var-EOF to achieve the analysis at each cycle, indicating better conditioning of the minimizations.

A significant reduction in RMSE (16%) was indicated by our results, achieved through the projection of the minimization onto the EOFs basis. Although a substantial number of EOFs is retained in the ensemble to preserve information, it is conceivable that computational time could be further reduced by decreasing the number of EOFs. However, it is important to note that the forward pass of the emulator and the computation of its adjoint are inherently performed in the  $(128 \times 128$  grid-cells) physical space. Consequently, the computational time savings from reducing the number of EOFs may not be substantial. On the other side, it might lead to a loss of the small scale information, which can also be observed in Fig. B2 ~~with the RMSE increase when, where the RMSE increases as~~ the truncation index ~~decrease~~ decreases, down to a certain threshold.

We introduce in Appendix E an inflation scheme to tune the background cost function term. Two versions are evaluated: a more often used constant model inflation and an adaptive one, based on the  $\chi_p^2$  ~~(with p degrees of freedom)~~ diagnostic (Michel, 2014). The adaptive background ~~error~~ inflation can be easily implemented in twin experiment scenarios, under a Gaussian noise simulation. We observe a modest improvement in the time-averaged RMSE, with most of the gains occurring at the beginning of each assimilation window. Interestingly, in both cases, we have to deflate the B-matrix for an optimal analysis

RMSE (around 0.5 to 0.6 with a seasonal dependency).

A major factor for the quality in ~~term-terms~~ of RMSE is the frequency of the observations. While ~~most of~~ the results above focus on a fixed number of observations per cycle, additional experiments using varying frequencies in a twin experiment setup ~~are detailed in Appendix ?? were presented in Fig. 5, showing important improvement in RMSE when increasing the frequency of the observations.~~ However, current satellite data either provide spatially and temporally sparse observations ~~(like Cryosat-2)~~ or smooth, time-averaged full coverage ~~products~~, which introduces inherent ~~time-correlations~~ correlations between ~~the observations errors (like CS2SMOS)~~. It is important to note that our comparison with neXtSIM-F occurs 9 days after the last batch of assimilated observations, thus falling outside the average observation window. Yet, in the case of real observations, there is currently no SIT retrieval dataset that provides daily, non-smoothed, and non-time-correlated SIT measurements ~~to validate our experiments against. Ongoing works are proposing ML-based approaches to derive complete daily sea ice freeboard fields from satellite altimetry at fine spatial resolution (5 km), by modeling the spatio-temporal covariance of daily fields rather than relying solely on temporal averaging (Gregory et al., 2024b; Chen et al., 2024).~~

Let us note that neXtSIM, as well as neXtSIM-F exhibits more ~~localized dynamics small-scale features~~ than our emulator, ~~which that~~ tends to smooth out the fields. By assimilating CS2SMOS observations, we lose small scale information. Yet, as the observations are extremely smooth, the comparison is not as fair to neXtSIM-F, which provides more small scale dynamics, and hence suffers from double penalty effects. Implementing a stochastic emulator (Finn et al., 2024a, b) could yield more physically consistent results by conserving spectral energy. This approach may enhance the performance of the model within the 4D–Var minimization framework, although it raises questions about the reliability of the associated gradients and the increase in computation time. Newer satellites will hopefully provide data at higher resolution and better accuracy, but we expect that the current trade-off between resolution (by altimeters) and coverage (by passive microwaves) will remain an issue

in the foreseeable future. Another option is to apply super-resolution algorithms to enhance local scale dynamics in the data assimilation system (Barthélémy et al., 2022). Exploring these possibilities could significantly improve the efficacy of data assimilation with emulators for sea ice modeling.

One important aspect to monitor is the evolution of the cost function during minimization, as well as its associated gradient. More details are provided in the Appendix C. As seen in Fig. C2, there is more than an order of magnitude difference in the gradient norm during minimization within a single cycle. It is worth noting that the only stopping criterion consistently achieved is related to the tolerance of the cost function, where its decrease becomes smaller than a given value. Occasionally, a second criterion related to the gradient norm may also be considered, which requires that the maximal value of the gradient for the entire field be below another specified threshold. However, this gradient criterion is never met in our case. As shown in Fig. C2, there are still some grid-cells, often located in the MIZ, where the gradient norm remains non-negligible. Yet, as 470 seen in Fig. C1, at each cycle, the cost function attains a stable minimum which indicate that the L-BFGS optimization worked correctly.

An important point to discuss is the realism of the emulator's adjoint. In deep learning, the high number of degrees of freedom often results in ~~poor-quality gradients~~ noisier gradients (Sitzmann et al., 2020). In the 4D–Var framework, this issue is mitigated by the background term, which regularizes the emulator's potentially noisy gradient. This noise is evident in the analysis, as shown in Fig. 3 and Fig. C2, particularly near the MIZ. While a dedicated training procedure for the emulator could potentially improve this aspect, it does not appear to hinder the current 4D–Var setup. In fact, the successful results achieved with this system indirectly validate the adjoint of both the emulator and the cost function. Furthermore, correctness checks for both adjoints are provided in Appendix C4.

Since the emulator provides fast forecasts of the SIT dynamics, and due to the fast data assimilation, we can investigate the possibility to run ensemble data assimilation (EDA) (Raynaud et al., 2008; Isaksen et al., 2010) by running an ensemble of 4D–Vars, with perturbations of the observations and the background term. This ensemble can be used to build an ensemble of trajectories, but also to improve the flow-dependency of the background covariances in the EOF space. Interestingly, this approach also allows for the comparison with other data assimilation methods using this emulator, such as the ensemble Kalman filter. This would enable a more straightforward comparison with current state-of-the-art data assimilation methods 485 for sea ice. However, it would be important to further discuss the capability of our deterministic surrogate model to generate a state ensemble in this context.

These results are promising and demonstrate the potential for using model emulators in data assimilation, particularly with classical methods in real-world applications. ~~However, the main challenge observed is the smoothness of both the emulator and the observations.~~ Furthermore, it could be interesting to see the impact of assimilating several variables, like SIT and sea-ice concentration (SIC) onto an emulator.

## 8 Conclusions

In this paper, we introduced the first 4D–Var system based on a surrogate model that is trained to fully emulate the evolution of the sea-ice thickness. This work is a preliminary step towards the use of fully emulated models in data assimilation. Through twin experiments, we initially demonstrate the ability of the surrogate model to leverage its automatic gradient in a 4D–Var 495 minimization. The 4D–Var system can be efficiently implemented by using EOFs extracted from the model’s climatology. The assimilation in EOF space improves the system compared to a diagonal background covariance. Inflation techniques bring small improvement. In the second part of the study, we investigate the assimilation of real observations with the 4D–Var system. Assimilating real observations improves the positioning of the MIZ. These observations act as a bias correction, highlighting the potential need for weak-constraint 4D–Var to address such biases. This could also create opportunities to train a bias correction 500 model or refine the emulator using analysis increments. With limited resources, such as emulating only sea-ice thickness and assimilating CS2SMOS observations, the developed 4D–Var system performs comparably to the ensemble Kalman filter-based operational neXtSIM-F system. Moreover, the emulator-based 4D–Var system is significantly more computationally efficient than ensemble Kalman filter systems relying on geophysical models. Although these results are derived from a coarse-grained emulator of the available neXtSIM dataset, no major issues are anticipated in increasing the resolution of both the surrogate 505 model and the observations. The computational cost of this data assimilation approach remains well within the standards of current sea-ice data assimilation systems.

## Appendix A: Surrogate modeling

In this section, we present the forecast ability of the emulator. The root-mean-squared error (RMSE) between the prediction

$\mathbf{x}_{n+k\Delta t}^f$  and the simulation  $\mathbf{x}_{n+k\Delta t}^t$  is computed over all pixels (i, j) of the field of size  $(N_x, N_y)$ ,

for each sample  $n$  of the validation set containing  $N_s = 1470$  trajectories, initialized at time  $t_n$ ,

$$\text{RMSE}(k) = \frac{1}{N_s} \sum_{n=1}^{N_s} \sqrt{\frac{1}{N_x \cdot N_y} \sum_{i,j}^{N_x, N_y} (\mathbf{x}_{t_n+k\Delta t}^f - \mathbf{x}_{t_n+k\Delta t}^t)^2} \sqrt{\frac{1}{N_x \cdot N_y} \sum_{i,j}^{N_x, N_y} (\mathbf{x}_{i,j}^f(t_n + k\Delta t) - \mathbf{x}_{i,j}^t(t_n + k\Delta t))^2}. \quad (\text{A1})$$

In order to quantify systematic errors of the surrogate model, we compute its mean error (bias). This metric tells about the ability of the neural network to correctly estimate the total amount of sea ice in the full domain,

$$\text{bias}(k) = \frac{1}{N_s} \sum_{n=1}^{N_s} \frac{1}{N_x \cdot N_y} \sum_{i,j}^{N_x, N_y} \left( \mathbf{x}_{i,j}^f(t_n + k\Delta t) - \mathbf{x}_{i,j}^t(t_n + k\Delta t) \right). \quad (\text{A2})$$

The code structure of the surrogate model  $g_\theta$  is presented in Alg. A1. Let us note that  $f_\theta$  maps  $\tilde{\mathbf{x}}_t$  to  $\tilde{\mathbf{y}}_{t+\Delta t} = \tilde{\mathbf{x}}_{t+\Delta t} - \tilde{\mathbf{x}}_t$  which corresponds to the normalized difference in sea-ice thickness over 12 hours. The normalization is defined as in Eq. (1) with  $\mu_{\text{out}}$  and  $\sigma_{\text{out}}$  the associated global mean and standard deviation of  $\mathbf{y}_{t+\Delta t} = \mathbf{x}_{t+\Delta t} - \mathbf{x}_t$ , computed over the training dataset.

---

**Algorithm A1** Full-state surrogate model  $g_\theta$  mapping from  $\tilde{\mathbf{x}}_t$  to  $\tilde{\mathbf{x}}_{t+\Delta t}$  using the previously trained  $f_\theta$ . This algorithm describe exactly how the state  $\tilde{\mathbf{x}}_{t+\Delta t}$  is obtained by the application of the fine-tuned  $f_\theta$  neural network as defined in Eq (2b).

---

**Require:**  $f_\theta(\tilde{\mathbf{x}}_t, \mathbf{F}, \theta)$ ,  $\tilde{\mathbf{x}}_t$ ,  $\mathbf{F}$ ,  $\sigma$ , and normalization values  $(\mu_{\text{SIT}}, \sigma_{\text{SIT}}, \mu_{\text{out}}, \sigma_{\text{out}})$

```

 $\tilde{\mathbf{y}}_{t+\Delta t} \leftarrow f_\theta(\tilde{\mathbf{x}}_t, \mathbf{F}, \theta)$ 
 $\mathbf{y}_{t+\Delta t} \leftarrow \sigma_{\text{out}} \tilde{\mathbf{y}}_{t+\Delta t} + \mu_{\text{out}}$ 
 $\mathbf{x}_t \leftarrow \sigma_{\text{SIT}} \tilde{\mathbf{x}}_t + \mu_{\text{SIT}}$ 
 $\mathbf{x}_{t+\Delta t} \leftarrow \mathbf{x}_t + \mathbf{y}_{t+\Delta t}$ 
 $\tilde{\mathbf{x}}_{t+\Delta t} \leftarrow \frac{\mathbf{x}_{t+\Delta t} - \mu_{\text{SIT}}}{\sigma_{\text{SIT}}}$ 
 $\tilde{\mathbf{x}}_{t+\Delta t} \leftarrow \sigma(\tilde{\mathbf{x}}_{t+\Delta t}) = g_\theta(\tilde{\mathbf{x}}_t)$ 
 $\tilde{\mathbf{x}}_{t+\Delta t} \leftarrow \text{ReLU}(\tilde{\mathbf{x}}_{t+\Delta t}) = g_\theta(\tilde{\mathbf{x}}_t)$ 

```

---

The training of the emulators are shown in Fig. A1, with the display of the training losses and the validation losses. The training loss measures how well the emulator fits the training data, while the validation loss assesses its performance on unseen

data to detect overfitting and ensure generalization. The baseline consists of the UNet trained by learning  $f_\theta$ . By transfer learning,  $f_\theta$  weights are fine-tuned in order to learn  $g_\theta$ , with a constrain ( $\lambda = 10$ ) inside the loss function, see Eq. (5). Results in term of forecast ability of those emulators are presented in Fig. A2. The forecast skill of  $g_\theta$  is compared to the one of  $f_\theta$ , and the persistence, which consist to take the initial condition as the constant state of the system. We can see that in terms of RMSE,  $g_\theta$  is slightly worse than the baseline, but in terms of bias, the fine-tuned constrained emulator display a smaller bias error compared to  $f_\theta$ .

**Figure A1.** Left: Training and validation losses of the surrogate model. Right: Training and validation global losses of the surrogate models. Lines Light blue and light green lines in transparency indicate the validation losses. The green dashed line indicates the experiment where the weights of  $f_\theta$  are frozen and a linear activation function is applied after the renormalization process. Blue line corresponds to the training of  $g_\theta$ . The weights of  $f_\theta$  are retrained with the new learning objective, with  $\lambda = 10$ .

**Figure A2.** Left: Forecast skill (RMSE) of the surrogate model. Right: Bias errors of the surrogate models. The green dashed line indicates the results for  $f_\theta$ . Blue line corresponds to the training of  $g_\theta$ . The weights of  $f_\theta$  are retrained with the new learning objective, with  $\lambda = 10$ .

## Appendix B: Empirical Orthogonal Functions

### B1 EOFs definition

We build an ensemble of perturbations using neXtSIM simulation outputs,  $\mathbf{X}$  of size  $\mathbb{R}^{N_t \times N_z}$ , with  $N_z = 8871$  the number of unmasked pixels and  $N_t$  the number of state in the ensemble, this number depends on the number of years taken to compute the ensemble and varies from 1500 to 11000. After the removal of the temporal mean from this ensemble:

$$\tilde{\mathbf{X}} = \mathbf{X} - \bar{\mathbf{x}}, \quad (B1)$$

we can compute the singular value decomposition of  $\tilde{\mathbf{X}}$ ,

$$\tilde{\mathbf{X}} = \mathbf{U} \Sigma \mathbf{V}^\top, \quad (B2)$$

with  $\mathbf{U}$  an orthogonal matrix of size  $(N_t \times N_t)$ ,  $\Sigma$  a diagonal matrix of size  $(N_t \times N_z)$  containing the  $N_t$  singular values of  $\tilde{\mathbf{X}}$  and  $\mathbf{V}$  an orthogonal matrix of size  $(N_z \times N_z)$ . We then define the EOFs  $\varphi$  as the columns of  $\mathbf{V}$ .

One advantage of using EOFs is the ability to reduce the dimensionality of the minimization space by projecting the state vector onto a truncated set of EOFs. We denote this truncation by  $\varphi_m$ , where  $m$  represents the truncation index. In practice, this involves limiting the projection of the orthonormal matrix to the first  $m$  EOFs, ordered by explained variance, thereby reducing the computational burden while retaining the most significant modes of variability. The four predominant EOFs are displayed in Fig. B1, as well as their associated variance.

### B2 Choice of the truncation index $m$

In order to validate the best truncation index, we run an experiment with 4D–Var-EOF, in the twin experiment setup, with a value of  $m$  ranging from 10 to 8871. We then compute the total RMSE over all cycles. Results are presented in Fig. B2. We observe that for  $m > 5000$ , the RMSE has reached a minimum. Let us note that using a smaller value of  $m$  reduces the minimization time, as it is reducing the dimension of the minimization space. Based on these results, and while we wanted to maintain a good reconstruction capacity, we chose to maintain  $m = 7000$  for all experiments.

## Appendix C: 4D–Var optimization

### C1 4D–Var algorithms

We present here the algorithms for the computation of the 4D–Var optimization, in the 4D–Var-EOF case. The computation of the cost function, for a given cycle, is shown in Alg. C1. The state  $\mathbf{w}_0$  is mapped back to the physical space, forecasted throughout the DAW using the emulator, where the observation term of the cost function is computed and then transformed back into the affine space of the EOFs to compute the observation term of the cost function. The total computation across all DAW is presented in Alg. C2.

### Four predominant EOFs

**Figure B1.** Four predominant EOFs of SIT, at the top left of each EOF is indicated the associated explained variance.

**Figure B2.** Average of the RMSE between  $\mathbf{x}_a$  and  $\mathbf{x}_t$  across all cycles and all timesteps for different values of  $k$  with Gaussian noise for the observations.

---

**Algorithm C1** Cost function  $(\mathcal{J})$  computation for the 4D–Var-EOF

---

**Require:**  $g_\theta, \mathbf{w}_0, \tilde{\mathbf{y}}_{k:1,\dots,K}$ , the number of application of the emulator between two observations  $N_f, K, \sigma_{\text{obs}}, \lambda_{\text{inf}}, \mathbf{H}, \varphi_m, \mathbf{F}$

$$\mathcal{J}_b = \frac{1}{2\lambda_{\text{inf}}^2} (\mathbf{w}_0 - \mathbf{w}_0^b)^\top (\mathbf{w}_0 - \mathbf{w}_0^b)$$
$$\mathcal{J}_o = 0$$
$$\tilde{\mathbf{x}}_0 = \bar{\mathbf{x}} + \varphi_m \mathbf{w}_0$$

**for**  $i$  in range  $(1, \dots, K)$  **do**

$$\tilde{\mathbf{x}}_i = g_\theta^{N_f}(\tilde{\mathbf{x}}_{i-1}, \mathbf{F}_{i-1 \rightarrow i}, \theta) \quad \tilde{\mathbf{x}}_i = g_\theta^{N_f}(\tilde{\mathbf{x}}_{i-1}, \mathbf{F}_{i-1 \rightarrow i})$$
$$\mathcal{J}_o = \mathcal{J}_o + \frac{1}{2\sigma_{\text{obs}}^2} \|\tilde{\mathbf{y}}_i - \mathbf{H}\tilde{\mathbf{x}}_i\|^2$$

**end for**

$$\mathcal{J} = \mathcal{J}_o + \mathcal{J}_b$$

---

---

**Algorithm C2** Wrapper for cycling the 4D–Var-EOF minimization, using the loss  $(\mathcal{J})$  defined in Alg. C1.

---

**Require:**  $g_\theta(\tilde{\mathbf{x}}_t, \mathbf{F}), \tilde{\mathbf{y}}_{k:1,\dots,K}, N_f, N_{\text{cycle}}, K, \mathbf{w}_0, \text{DOF}, \lambda_{\text{inf}}, \varphi_m$

**for**  $n$  in range  $(1, \dots, N_{\text{cycle}})$  **do**

$$\mathbf{w}_{0,n}^a = \mathbf{L} - \text{BFGS}(\mathbf{w}_{0,n}, \text{loss}(\mathbf{w}_{0,n}, \tilde{\mathbf{y}}_{k:1,\dots,K}))$$
$$\tilde{\mathbf{x}}_{0,n}^a = \bar{\mathbf{x}} + \varphi_m \mathbf{w}_{0,n}^a$$
$$\tilde{\mathbf{x}}_{N_f(n \rightarrow n+1)}^a = g_\theta(\tilde{\mathbf{x}}_{0,n}^a, \mathbf{F}_{n \rightarrow n+1})$$
$$\tilde{\mathbf{x}}_n^b, \tilde{\mathbf{x}}_n^0 = \tilde{\mathbf{x}}_{N_f(n+1)}^a$$
$$\mathbf{w}_{0,n+1} = \varphi_m^\top (\tilde{\mathbf{x}}_n^0 - \bar{\mathbf{x}})$$
$$n \leftarrow n + 1$$

**end for**

---

The L-BFGS-B (Broyden, 1967; Liu and Nocedal, 1989) algorithm is used to minimize the cost function  $\mathcal{J}$ , defined in 555 Alg. C1. The optimization is constrained by bounds defined over all the variables of  $\tilde{\mathbf{x}}$  with a minimal value set to  $\text{min\_SIT}$   $\text{SIT}_{\text{min}}$  as defined previously, in the case of the 4D–Var-diag. Two criteria are used to stop the minimization:  $f_{\text{tol}}$ , which corresponds to a threshold below which the cost function improvement is considered sufficient, and a gradient norm threshold  $g_{\text{tol}}$ , below which the norm of the gradient must fall. Note that we did not define a maximal number of iterations of the L-BFGS-B and the criteria  $f_{\text{tol}}$  was systematically reached.

In practice, in our case the only stopping criterion used is  $f_{\text{tol}}$ , as the gradient is significantly decreasing, yet, some instabilities at each iteration, especially on the MIZ are still observed.

## C2 Cost function analysisdiagnostics

The value of the cost function  $\mathcal{J}$  as defined in Eq. (6b) or Eq. (9) and minimized with a L-BFGS-B can be followed across all cycles, as shown in Fig. C1. During the first cycles, the seasonality The RMSE increase observed in Fig. 4 can also be 565 observed in the cost function. The increase in May comes after 9-10 cycles, which corresponds to 2 cycles before the time

**Figure C1.** Cost function minimization with L-BFGS optimizer across all cycles for 4D–Var-EOF with log-normal noise, the total cost function is shown in blue, this term can be decomposed with the background loss term (green term) and the observation loss term (orange curve). Note that the y-axis is in log scale.

where the observation cost function decreases and the background cost function increases and becomes predominant. Based on this observed seasonality, we introduce an adaptive background strategy to offer an estimation of the order of magnitude of the background cost function. This implementation is further described in SeeAppendix. E.

### C3 Gradient analysis

We can also investigate the gradient of the cost function, and its evolution between the beginning and the end of the DAWcycle, as shown in Fig. C2. We can observe a global decrease of the gradient across the full Arctic. Yet, some grid-cells keep a strong gradient, especially on the MIZ.

### C4 Tests of the adjoint of the emulator and the cost function gradients

First of all, we test the gradient of the cost function. With Taylor's expansion

$$\mathcal{J}(x + \epsilon h) = \mathcal{J}(x) + \epsilon \mathcal{J}'(x) \cdot h + \mathcal{O}(\|\epsilon\|^2), \quad (\text{C1})$$

we look at the ratio

$$\mathcal{I}(\epsilon) = \left\langle \left| \frac{\mathcal{J}(x + \epsilon h) - \mathcal{J}(x - \epsilon h)}{2\epsilon \mathcal{J}'(x) \cdot h} \right| \right\rangle_{\|h\|=1}, \quad (\text{C2})$$

with  $x$  a state on the trajectory of the emulator. Note that we have

$$\mathcal{J}(x + \epsilon h) = \mathcal{J}(x) + \epsilon \mathcal{J}'(x) \cdot h + \mathcal{O}(\|\epsilon\|^2) \quad (\text{C3a})$$

$$\mathcal{J}(x - \epsilon h) = \mathcal{J}(x) - \epsilon \mathcal{J}'(x) \cdot h + \mathcal{O}(\|\epsilon\|^2). \quad (\text{C3b})$$

By making the difference of those two equations we obtain

$$\mathcal{J}(x + \epsilon h) - \mathcal{J}(x - \epsilon h) = 2\epsilon \mathcal{J}'(x) \cdot h + \mathcal{O}(\|\epsilon\|^2). \quad (\text{C4})$$

**Figure C2.** Evolution of the gradient under minimization during the 4D–Var 4D–Var-diag minimization. In The left panel is displayed shows the gradient at the end of the first minimization and on of one cycle (cycle 31), while the right panel is outlined displays the gradient at the end of the last minimization of the same cycle, when the minimization criterion is reached. In this case, the synthetic observations are perfect observations (no noise addition).

By averaging, we expect

$$\mathcal{I}(\epsilon) \simeq 1 + \mathcal{O}(\|\epsilon\|). \quad (\text{C5})$$

In practice, we take for the trajectory the full 2 years free run without assimilation, with the same parameters as defined in Sec. 5, with observations perturbed with cond-clipped noise. We evaluate several values  $\mathbf{h}$ , including canonic vectors to focus only on single pixel, especially in the MIZ, and in Central Arctic. We define the canonic vectors  $\mathbf{h}_{\text{zone}} \in \mathbb{R}^{8871}$  as

$$\mathbf{h}_{\text{zone}} = (0, \dots, 0, 1, 0, \dots, 0), \quad (\text{C6})$$

with the 1 at the position corresponding to the chosen area. We select 5 pixels for the MIZ and 5 pixels for Central Arctic. The results are shown in Fig. C3. While the residual errors remain small, they increase for  $\epsilon 

**Figure C3.** Logarithm of the absolute value of  $\mathcal{I}(\epsilon)$  for several values of  $\epsilon$ . Black line corresponds to the choice of a random perturbation of the cost function, with 10 experiments performed. Red line corresponds to a perturbation inside the Central Arctic region, with 5 experiments performed. Green line corresponds to a perturbation in the MIZ, with 5 experiments performed. Blue line corresponds to the expected evolution identity  $\mathcal{O}(\|\epsilon\|)$ .

We study the ratio

$$\mathcal{I}_{\mathbf{z}}(\epsilon) = \left\langle \left| \frac{\mathcal{L}(\mathbf{z}, \mathbf{x} + \epsilon \mathbf{h}) - \mathcal{L}(\mathbf{z}, \mathbf{x} - \epsilon \mathbf{h})}{2\epsilon \mathbf{z}^\top \mathbf{M}'(\mathbf{x}) \cdot \mathbf{h}} \right| \frac{\mathcal{L}(\mathbf{z}, \mathbf{x} + \epsilon \mathbf{h}) - \mathcal{L}(\mathbf{z}, \mathbf{x} - \epsilon \mathbf{h})}{2\epsilon \mathbf{z}^\top g'_\theta(\mathbf{x}) \cdot \mathbf{h}} \right\rangle_{\|\mathbf{h}\|=1}, \quad (\text{C9})$$

with  $\mathbf{x}$  in the trajectory of the emulator. We expect the same behavior as for  $\mathcal{I}(\epsilon)$ . In this case, we look at several values for  $\mathbf{z}$ , defined exactly as the previous  $\mathbf{h}$ . The emulator is evaluated onto the full 2017-2018 dataset. Results are presented in Fig. C4. We obtain satisfying results, with a divergence of the residual for  $\epsilon 

**Figure C4.** Logarithm of the absolute value of  $\mathcal{I}(\epsilon)$  for several values of  $\epsilon$ . Black-Blue line corresponds to the choice of a random perturbation of the cost function, with 10 experiments performed. Orange line corresponds to a perturbation inside the Central Arctic region, with 5 experiments performed. Green line corresponds to a perturbation in the MIZ, with 5 experiments performed. Blue-Red line corresponds to the expected evolution identity  $\mathcal{O}(\|\epsilon\|)$ .

**Figure D1.** 2017 and 2018 RMSE of 4D-Var-EOF depending on the number of observations in every assimilation cycle the twin experiment case, with cond-clipped perturbations. Curves indicated The curve represents the RMSE of the 4D-Var analysis with regard to neXtSIM SIT, with 2 observations per cycle (purple curve), 4 observations per cycle (blue curve), 8 observations per cycle (gray curve) and 16 observations per cycle (green curve).

## Appendix E: Background inflation

We adopt an adaptive multiplicative inflation scheme on the background error term, materialized by  $\lambda_{\text{inf},n}$  and which is evalu-

ated on each cycle  $n$ . It can be decomposed in two terms:

$$\lambda_{\text{inf},n} = \lambda_m \times \lambda_{a,n}. \quad (\text{E1})$$

$\lambda_m$  corresponds to the inflation term associated with the model error. It is constant throughout the experiments. In order to test the consistency of the solution of the innovation vector, we perform the  $\chi^2_p$  test. Under Gaussian assumption, the minimum

value of the cost function has a  $\chi^2_p$  distribution with  $p$  the number of observations assimilated (Michel, 2014). It means that the average of the cost function minimum should stay around  $p$ , in our case  $8871 \times N_{\text{obs}}/2$ , with  $N_{\text{obs}}$  the number of observations per window. Yet, with the different noise investigated in our study, this test does not work under other assumptions.

Following the  $\chi^2$  assumption, for each cycle, the minimal value of the cost function  $\mathcal{J}$  should in average be equal to the number of degree of freedom (DOF), which is equal to the total number of observations  $p$ . In practice, we define for each cycle  $n$ ,

$$\lambda_{a,n+1} = \sqrt{\frac{\mathcal{J}_b^n}{|\text{DOF} - \mathcal{J}_o^n|}}, \quad (\text{E2})$$

where  $\mathcal{J}_b^n$  corresponds to the background term of the cost function at cycle  $n$  and  $\mathcal{J}_o^n$  corresponds to the observation term of the cost function at cycle  $n$ ; with for the initial cycle, a value of 1 for  $\lambda_{a,0}$ . The multiplication of the two terms  $\lambda_m$  and  $\lambda_{a,n}$  gives us the total inflation  $\lambda_{\text{inf},n}$  in the adaptive case.

In order to help the optimization of the 4D–Var, we investigate the use of inflation scheme as defined in Eq. (E2). Two schemes are evaluated, a constant inflation, only modeled by  $\lambda_m$  and an adaptive inflation scheme with the multiplicative

addition multiplication of  $\lambda_a$  based on the  $\chi^2$  estimation of  $\mathcal{J}$  and  $\lambda_m$ . Only Gaussian noise for the observations are considered

and the value of  $\lambda_m$  ranges between 0.6 and 3. Results with the mRMSE are outlined in Fig. E1. As observed before we can see that 4D–Var-EOF is clearly better than 4D–Var-diag 4D–Var-diag, but both inflation schemes have similar behavior in both cases. Using an inflation scheme ( $\lambda_{\text{inf}} \neq 1$ ) yields better results in terms of mRMSE in both cases if we correctly tune  $\lambda_m$ .

Interestingly, in the case of constant inflation, the mRMSE increases almost linearly with  $\lambda_m$ . This indicates that, when using constant inflation, emphasizing the  $\mathcal{J}_b$ – $\mathcal{J}_o$  term tends to undermine the performance of the 4D–Var. Conversely, in the adaptive case, assigning a significant value to  $\lambda_m$  while still optimizing the ratio between  $\mathcal{J}_b$  and  $\mathcal{J}_o$  results in improved mRMSE.

As shown in Fig. E2, in which is plotted the ratio between the adaptive inflation scheme with  $\lambda_m$  set to 3 and the run with Gaussian noise without any inflation scheme, in 2018, we can see that the benefit from inflation is primarily observed at the start 640 of the DAW and more prominently at the beginning of the year. On average, the gain from the adaptive inflation is mitigated, as some end of cycle ratio are above 1. By using the inflation, we favor the improvement at the beginning of the DAW to the detriment of the end of the DAW. Let us also note that the major gain from the inflation come from March to July, which corresponds to the period where the RMSE of the analysis  $x_a$  is higher, hence, where the 4D–Var is struggling the most.

**Figure E1.** mRMSE of  $x_a$  for both 4D-Var-EOF (dark blue lines) and 4D-Var-diag (green lines), for the two types on background inflation proposed, the constant inflation (dashed lines) and the adaptive inflation (solid lines). The results are plotted as a function of the model inflation  $\lambda_m$ . In the case of the constant inflation,  $\lambda_a$  is automatically set to 1 whereas in the adaptive case, it is defined as per Eq. (E2).

**Figure E2.** Evaluation of the gain from the adaptive inflation, compared to no inflation, for both 4D-Var-EOF (dark blue lines) and 4D-Var-diag (green lines). The ratio between the RMSE of  $x_{a,\text{inflation}}$  and  $x_{a,\text{noinflation}}$  is plotted with respect to time. The dotted red line indicates the position where there is no gain from the inflation. A value of this ratio below 1 means a gain from the adaptive scheme whereas a value above 1 corresponds to a loss in RMSEs.

To conclude, adaptive background inflation can be easily implemented in twin experiment scenarios. We observe a modest  
improvement in the total average RMSE, with most of the gains occurring at the beginning of the assimilation windows. In the  
case of adaptive inflation, the inflation values fluctuate around 0.5, exhibiting a seasonal pattern.

*Code and data availability.* The outputs of neXtSIM model used here (Boutin et al., 2022) are available at [https://ige-mecom-opendap.univ-grenoble-alpes.fr/thredds/catalog/meomopendap/extract/SASIP/model-outputs/OPA-neXtSIM\\_CREG025/OPA-neXtSIM\\_CREG025-ILBOXE140catalog.html](https://ige-mecom-opendap.univ-grenoble-alpes.fr/thredds/catalog/meomopendap/extract/SASIP/model-outputs/OPA-neXtSIM_CREG025/OPA-neXtSIM_CREG025-ILBOXE140catalog.html). Forcings data from ERA5 are publicly available in the Copernicus Data Store (C3S, 2018). All the codes to build the datasets,  
train the emulator and build the 4D-Var system are provided, the jupyter-notebook used to create the figures, as well as the post-processed datasets and neural network weights (Durand, 2024). Additionally, for real observations assimilation, CS2SMOS retrievals are publicly

available (European Space Agency, 2023). ECMWF forecast (Owens and Hewson, 2018) were provided by Alban Farchi from the ECMWF. neXtSIM-F forecast archives are not publicly available but can be requested by e-mail from [nextsimf@nersc.no](mailto:nextsimf@nersc.no).

*Author contributions.* CD, TSF, AF, and MB refined the scientific questions and prepared an analysis strategy. JB and LB provide technical  
knowledge on sea-ice data assimilation. CD performed the experiments. CD, TSF, AF, and MB analyzed and discussed the results. CD wrote the manuscript with TSF, AF, MB, JB, and LB reviewing.

*Competing interests.* The authors declare that they have no conflict of interest.

*Acknowledgements.* The authors acknowledge the support of the project SASIP (grant no. *G* – 24 – 66154) funded by Schmidt Sciences – a philanthropic initiative that seeks to improve societal outcomes through the development of emerging science and technologies. This  
work was granted access to the HPC resources of IDRIS under the allocations 2021-AD011013069, 2022-AD011013069R1, and 2023-AD011013069R2 made by GENCI. The authors would like to thank Timothy Williams for its availability and access to neXtSIM-F forecast. Cerea is a member of the Institut Pierre-Simon Laplace (IPSL). This manuscript was grammatically revised using ChatGPT.

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
