# Peer review of "Four-dimensional variational data assimilation with a sea-ice thickness emulator"

_EGUsphere, 2024_

## Referee Comment (RC2)

Review of "Four-dimensional variational data assimilation with a sea-ice thickness emulator"

by Charlotte Durand et al.

Version 1

Date of review: 20 March 2025

**Summary**

This manuscript assesses the performance of 4D-Var data assimilation of sea-ice thickness (SIT) in the context of an SIT emulator featured in a recent publication of *The Cryosphere* (DOI: 10.5194/tc-18-1791-2024), taking advantage of the easy availability of the emulator's adjoint. It is a scientifically interesting development that, I believe, is worth having its place in the scientific literature. Therefore, having considered the manuscript's contents in relation to the remits of *The Cryosphere*, I recommend its publication in the journal in principle. That being said, I do have major reservations about the manuscript in its current form, and I think it needs to be substantially revised before publication. The reasons are given in the "Major Comments" section below and elucidated in more detail in the "Minor Comments" section. I understand the list of minor comments is very long, but I hope they could provide concrete suggestions on how the manuscript could be improved, which will also help address the major comments.

**Major Comments**

1. Overall, I find the Introduction (Section 1) a bit fragmented and lacking a general theme or direction that the authors would like to guide the reader towards.

2. For better clarity, I recommend a reorganisation of the manuscript, to avoid the reader having to go back and forth between the idealised-observation experiments and the CS2SMOS experiments. My recommendation is as follows. After the introductory section, the model, the emulator, the two flavours of 4D-Var and the metrics used for evaluation would be presented first (possibly in separate sections). Next, there would be two main sections, one about the idealised-observations experiments, and the other about the CS2SMOS experiments. In each of these sections, you could first describe the setup of the observations before moving on to discuss the experimental results. Following these sections, some discussion relevant to both sets of experiments could be presented in the final section, where concluding remarks could also be given.

3. I think there is too much content in the appendices. Some of the contents are important and should sit in the main text (see e.g. Minor Comments 47 and 104). Other parts of the appendices (such as Appendix E) could be scientifically interesting, but I don't see a good reason that the material should be included in the first place (cf. Minor Comment 72). There should be good coherence between the main text and appendices.

4. A recurring theme in the minor comments below is that results or claims may not be adequately explained or backed by evidence. Generally speaking, statements need to be

more qualified, and caveats need to be mentioned. This is to avoid the impression that the results are over-generalised.

5. Another recurring theme in the minor comments below concerns the language and presentation. Despite the manuscript having been grammatically edited with ChatGPT (cf. line 555), there remain a number of grammatical errors and many instances where the usage of the English language could be improved. Some abbreviations and symbols have not been defined at their first use, and some parts of the text may sound a bit cryptic to a non-specialist audience. Please revisit the text and make it more accessible to a general audience.

**Minor Comments**

1. Line 21: Please explain what neXtSIM is.
2. Line 24: CryoSat should be CryoSat-2; SMOS has not been defined.
3. Line 25: What does the "-F" refer to?
4. Lines 26 – 32: You open the paragraph using the term "data assimilation techniques" and later you mention about nudging. However, I wouldn't call nudging a DA technique. Perhaps "initialisation techniques" would be a better term.
5. Line 27, regarding the word "predominantly": I am not sure if EnKF is the most popular way of initialising sea ice in forecasting systems. Variational methods are also common.
6. Line 41 – 42: I think the part of the sentence "as well as a background term… state of the system" is a distraction. A background is required for all DA methods, so this is not something special about 4D-Var, yet the way the sentence is worded seems to suggest that this is a characteristic feature of 4D-Var.
7. Line 43: The word "at" is unnecessary.
8. Lines 44 – 45: The use of "to" at the beginning of the sentence seems to suggest that the propagation of gradient information backwards is the aim. However, I think the aim is what you have said in the next sentence instead: to allow information from later parts of the DAW to be incorporated in the analysis at the beginning of the DAW. I think the sentence could be reformulated in a clearer and more concise way.
9. Line 45: Why is the minimisation's dependence on the adjoint implicit?
10. Line 47: It would be nice to mention the adjoint of NAOSIM (see DOI: 10.1029/2008GL036323).
11. Lines 48 – 49: You say that adjoint models yield limited realism for sea-ice simulations. Yet, as things stand, the remaining sentences in this paragraph don't elaborate on this. I would like to see this explained.
12. Lines 52 – 53: My understanding is that the adjoint was already available before its incorporation into MIT-GCM.
13. Section 2.1, section heading: This section includes description about adaptations of the neXtSIM grid and model variables to the neural network model you are going to use. Perhaps a section heading reflecting this would be more accurate.
14. Lines 81 – 82: The abbreviation NEMO OPA is undefined.

15. Line 91: Could you explain why a normalised variable has to be used as opposed to the original variable?

16. Line 93: When you talk about global statistics (both mean and standard deviation), is the computation of these statistics performed after the coarse-graining, i.e. at the $128 \times 128$ resolution?

17. Section 2.1, final paragraph: It may be more appropriate to move this discussion to Section 3 when the surrogate model is described, especially when the forcing vector $\mathbf{F}$ is mentioned in that section.

18. Line 99: Is the "Eulerian curvilinear grid" the coarse-grained grid or the finer grid?

19. Lines 105 and 190: Please specify units for the observation-error variance.

20. Line 108, the first part of Equation 2b: As far as I understand, the tilde quantities are normalised quantities, so shouldn't the equation be $\tilde{\mathbf{x}}_{t,\text{obs}}^{\text{LN}} = (\mathbf{x}_t \exp(\dots) - \mu)/\sigma$ instead?

21. Line 109: The notation $\min_{\text{SIT}}$ is confusing. With SIT in the subscript, I thought, at first sight, that it meant the minimum of something over SIT, with the argument (the "something") lacking.

22. The sentence spanning over lines $110 - 111$: There are probably too many unnecessary commas that hinder my ability to understand this sentence.

23. Line 113: You say that a variant to log-normal noise is introduced in Equation 2c, but where is the log in the equation?

24. Section 2.2.2, opening paragraph: A short elaboration about the complementarity of CryoSat-2 and SMOS observations would be desirable here.

25. Line 125: What is "Kriging"?

26. Line 132: Can you provide a graph of the function described in Equation 3, for easier visualisation? Please also define the units used in the coefficients for this equation.

27. Lines $134 - 136$: This is not a problem, because you do this in a pre-processing step. The assimilated "observations" are actually retrievals.

28. Lines $136 - 137$: Do you thin the observations or create super-observations? If not, how confident are you that the observation errors are negligibly correlated?

29. Lines $144 - 147$: How do you handle the non-differentiability of the ReLU function at the kink (in your case, at $\min_{\text{SIT}}$)?

30. Line 150: Is $N_x \times N_y$ the same as the $128 \times 128$ grid you mentioned earlier? Are the masked cells (due to land) excluded?

31. Line 159: Why is the clipping in Equation 4b omitted during the training?

32. Line 160: Is there a reason for using such a big weight at the initial training stage, but not at the second stage?

33. Section 4.1: It would be nice to mention somewhere that this is full 4D-Var (if I understood correctly) instead of the incremental 4D-Var that is commonly used in NWP centres. It took me (as a reviewer) a while to figure out that the 4D-Var experiments here are conducted in full-field space instead of incremental space.

34. Lines $169 - 170$ and 505: Presumably multiple observations valid at the same time are assimilated. How many observations are used at each observation time? The use of "$k^{\text{th}}$ observation" should be avoided if you mean the vector of observations at the $k^{\text{th}}$

observation time. Similarly, in line 505, it should be the number of observing times per cycle instead of the number of observations per cycle.

35. Line 170 and beyond: If everything from this point onwards is in normalised space, then it would be desirable to drop the tilde notation.

36. Line 171: Without having read the following sub-sections, a reader would naturally wonder what the specifications for **B** are. It may be useful to mention here that such details are given in the following sub-sections.

37. Lines 173 – 174: Do you mean that the background for the first cycle is the neXtSIM "truth"? Idealised DA experiments should begin with a background that is the truth perturbed by random statistics of the **B** matrix, rather than the truth itself.

38. Line 174: You start the experiments before the training period was over. Are there any implications?

39. Line 176: What is the "that" referred to in "After that"? Also, do the "both cases" refer to 4D-Var-diag and 4D-Var-EOF?

40. Lines 176 – 177: Is the **H** matrix simply a matrix of ones and zeros, a sub-selection of rows of the identity matrix (depending on your observation coverage)? **H** won't be a diagonal (square) matrix unless you have exactly the same number of observations as the number of elements in your state vector.

41. Line 178: What is $N_{\text{cycle}}$?

42. Line 181: I wouldn't regard this as two "types" of 4D-Var. They only differ by the choice of the **B** matrix effectively. For the 4D-Var-EOF case, according to your description in lines 201 – 202, it is equivalent to having $\mathbf{B} = \boldsymbol{\varphi}_m \boldsymbol{\varphi}_m^{\text{T}}$.

43. Line 181: It is not appropriate to refer to **B** as the "background" matrix. It is a covariance matrix of background errors, not of the background itself. Similarly, you are inflating the background errors, but not the background, in lines 191 and 340, 507 and 508.

44. Lines 192 – 193: Generally speaking, inflating the observation-error covariances is not equivalent to inflating background-error covariances. This is because the **B** matrix in 4D-Var is implicitly propagated in time by the linearised model **M** (in order words, **B** becomes $\mathbf{MBM}^{\text{T}}$). I would like to see how the covariances or correlations look like when **B** is implicitly propagated by the linearised model **M**.

45. Lines 199 – 200: When you say "The projection onto the EOFs enables access to cross covariances", I imagine you say this as an advantage compared to the 4D-Var-diag case where there is no cross-covariance. However, the diagonal **B** case is quite extreme and impractical, so I wouldn't say this as a feature of using EOFs. You also suggest that the projection onto EOFs improves numerical conditioning. Yet, since the **B** matrix for the 4D-Var-diag case is a scalar multiple of the identity matrix, its condition number is 1 and therefore the numerical conditioning cannot be improved beyond that.

46. Line 208: It is better to express the second term of the cost function in terms of $\mathbf{w}_0$.

47. Line 210: I think you should state the choice of the truncation index here. This is important information that shouldn't be relegated to an appendix.

48. Line 222: Is there a reason to average the RMSE over all grid points but keep it as a function of time? Unless there is a good reason to look at the results for individual assimilation cycles, I think it would be more interesting to evaluate the RMSE by taking

the mean over assimilation cycles yet presenting the results as a function of spatial location (i.e. map plots).

49. Lines 228 – 232: IIEE is standard terminology. If you use a modified definition, it is better to give an alternative name to it, to avoid being misleading. Do you have any evidence demonstrating that an SIT threshold of 0.1 m is roughly equivalent to an SIC threshold of 0.15? In any case, it would be good to show results after weighing them by grid cell areas, especially when the grid cell areas are not uniform.

50. Line 235: What do you mean by "the same past field"?

51. Line 237: For the cycle that spans from late 2017 to early 2018, is the result discarded?

52. Line 238: "45 cycles" means 720 days, so the last cycle should end on 21 December 2018 (end of the day), not 22 December.

53. Line 241 and caption of Table 1: Why would you like to assimilate perfect observations (taken from the truth)? If you do that, please define "the RMSEs between neXtSIM and the perturbed observations" mathematically, as you do with the other RMSE definitions.

54. Line 242: If the results are comparable and there is no discussion about the advantages and disadvantages of different noise types, then perhaps you don't have to show the results for all 3 types of perturbations.

55. Line 242: "type" should be in the plural.

56. Line 245: I think the improvement is more likely caused by the realism of the **B** matrix instead of the preconditioning and off-diagonal terms.

57. Figures 3 and 7: Some characters are missing in the legend of the graphs.

58. Figure 3: I am a bit surprised by the small difference between the grey and purple curves. Could you extend this graph to show days 0 to 15 as well? I would expect the difference to be larger at the beginning of the assimilation window, where the difference in the **B** matrix is most significant. In a sense, as the assimilation window goes on, the difference might diminish by the fact that it is the same linearised model **M** that propagates **B** (cf. Minor Comment 44).

59. Figure 3: The horizontal axis here is not an absolute date, but in terms of days after the start of the DAW. How do you compute the black dashed line in this case? Why is it not flat, but instead changes slightly with increasing lead time?

60. Line 250, regarding the words "we observe a slight improvement of 1.1 cm": Do you mean towards the end of the forecast when the red dashed line in Figure 3 is above the purple line? 1.1 cm seems to be minimal. Do you know whether the result is statistically significant? If so, how would you explain it (the 4D-Var-EOF case being better than the emulator forecast from perfect initial conditions)?

61. Lines 252 – 253: Do you know why the analysis is noisier than the background? Is it because the forecast (using the emulator) smooths out features?

62. Line 255: Is the negative bias a known issue of the surrogate model, or is it only an issue at this time of the year?

63. Figure 4: Is this for 4D-Var-diag or 4D-Var-EOF? If it is 4D-Var-diag, the only reason that you are not getting point-based increments is the implicit propagation of **B** by the linearised model **M** (cf. Minor Comment 44).

64. Caption of Figure 4:

a. Please state the full 16-day range of the assimilation window instead of just "2017-06-10".

b. It is not clear what the valid date / time of the "forecast" and "analysis" is.

c. Why do you choose to show the results of this cycle? You said earlier that you treat 2017 as a spin-up year.

65. Caption of Figure 5: How do you launch a new forecast every 6 hours when the assimilation cycle is 16 days long?

66. Line 256: There are exceptions to "the initial analysis RMSE [being] lower than the first guess RMSE" as indicated in the top panel of Figure 5. For example, in late February 2018, the analysis RMSE is larger than the background RMSE. Do you have any idea why that is the case?

67. Lines 258 – 259: I would say the peak is in June and not May, and I think it is difficult to conclude that the RMSE is rising again in December just by looking at the top panel of Figure 5.

68. Line 260: Without further evidence, I would be reluctant to conclude "4D-Var struggles more with dynamic shifts" only based on the coincidence between the trends in the top panel of Figure 5 and the seasonal changes in sea-ice extent. As you say in the next sentence, it could have something to do with the bias of the emulator.

69. Line 266: Do you use (northern hemisphere) winter because CryoSat-2 data are unavailable in the summer?

70. Lines 266 – 268: Would it be possible to randomly select a subset of CS2SMOS observations and not assimilate them, but rather use them for verification? This would give better confidence in the interpretation of results.

71. Lines 268 – 269: The date 10 October 2016 is a long way before the verification period (winter 2020 – 2021). Is there a reason to start so early? Also, this date is inconsistent with the date given in the caption of Figures 6 and 7.

72. Line 270: I note that the inflation of the **B** matrix is not used in both the idealised experiments and CS2SMOS experiments. In that case, does it need to be mentioned in the paper (whether in an appendix or not)?

73. Lines 271 – 273: I wonder how independent neXtSIM-F is compared to your neXtSIM-based emulator, given that both are related to neXtSIM. Could you elaborate on that?

74. Lines 274 – 275, "NeXtSIM-F assimilates… with a simple nudging": This is not clear. Are you assimilating or nudging CS2SMOS observations?

75. Line 275: Is the 9-day forecast from the start or end of the assimilation window?

76. Lines 276 – 277: If I understand correctly, you are using neXtSIM-F only as a reference (or "independent") gridded forecast dataset for verification purposes. Why do you have to compare neXtSIM-F against CS2SMOS observations?

77. Line 278: Could you elaborate on what "fairness" you refer to?

78. Line 279: "that are run" should be "that is run".

79. Line 279: Is there a reason to force the model with HRES instead of ERA5?

80. Lines 283 – 284: How difficult is it to find an initialisation field in 2020 for your winter 2020 – 2021 experiment? Could we get round this problem?

81. Line 286: "In average" should be "On average".

82. Lines 287 – 288: Do you know if there is anything that drives the trend in the difference between the experiment's 9-day forecast RMSE and the neXtSIM-F 9-day forecast RMSE?

83. Lines 288 – 289: Do you have evidence demonstrating the claim? If double penalty is the issue, it might be more suitable to use the Fractions Skill Score (see e.g. DOI: 10.1175/2007MWR2123.1).

84. Lines 290 – 291: More evidence is needed to support the claim that the DA system is "less efficient during periods of strong dynamic change". Perhaps you could consider running a multi-year experiment.

85. Lines 291 – 293: The sentence is not clear. Which are the "dynamic" periods you refer to?

86. Line 295: What "trajectory" do you mean?

87. Line 298, regarding the words "the bias systematically decreases": During the forecast stage (dashed purple lines in the top panel of Figure 7), the value of the bias indeed systematically decreases, but depending on whether the bias is above or below zero, it could mean either a worsening or an improvement. The wording needs to be more careful here.

88. Lines 298 – 299: I am not sure what you mean by "the emulator considers the analysis to increase the amount of sea ice".

89. Lines 299 – 300: Is there any evidence demonstrating your claim, namely that neXtSIM having a different SIT distribution from CS2SMOS could explain the phenomenon?

90. Line 301: Do the percentages refer to the IIEE at the end of the forecast, or at some other lead time?

91. Legend of Figure 6: The use of "ECMWF forecast" is misleading, as you are only using ECMWF fields to force your SIT emulator.

92. Caption of Figure 6: Please specify in the caption that the "RMSE values from neXtSIM-F" are RMSEs of 9-day forecasts.

93. Figure 7: The graphs are a bit messy. It is sometimes difficult to distinguish between solid purple lines and dashed purple lines.

94. Figure 8, top-right panel: The heading "Analysis" is misleading, as you say in the caption that it is a 9-day forecast from the analysis.

95. Caption of Figure 8: Are the CS2SMOS observations shown in the bottom-left panel of the figure also valid on 26 March 2021, or they span over a larger date range?

96. Line 320: "speculate" might be more appropriate than "infer" if it is not backed by evidence.

97. Line 323: It would be good to explain what "NVIDIA A100 SXM4 80 GB GPU" means. This string of letters and numbers seems a bit cryptic.

98. Line 325: What contributes towards the time for "gradient computation"? Apart from the running the adjoint of the emulator, is there anything else?

99. Lines 328 – 329: You say that "fewer iterations of the L-BFGS optimizer are required on average for 4D-Var-EOF to achieve the analysis at each cycle". Do you have evidence demonstrating this?

100. Lines 329 – 330: You mention about the conditioning of the minimisation. I would like to see how the dimensionality of the problem plays a role here, especially with small $m$ in the EOF case (see also Minor Comment 122). Reducing the number of EOFs, as you indicate in line 335, could reduce computational time by having fewer iterations in the minimisation before reaching convergence. (Theoretically, you should be able to reach the actual minimum by $m$ iterations, when $m \leq N_z = 8871$.)

101. Lines 336 – 337: It is true that the RMSE increases when $m$ decreases. However, according to Figure B2, $m = 3000$ and $m = 7000$ don't make a significant difference in RMSE.

102. Line 339: It is not clear that the notation $\chi_p^2$ refers to the chi-squared distribution / test with $p$ degrees of freedom.

103. Line 344: "in terms of" instead of "in term of".

104. Lines 345 – 346:
    a. I think changing the observation frequency is scientifically interesting, so the material shouldn't be relegated into an appendix. I would suggest that the additional experiments and results be discussed in more detail in the main text.
    b. It is better to use "observation frequencies" instead of just "frequencies", as the latter could mean wave frequencies (it doesn't make sense in the current context but could still create confusion for the reader). See also Minor Comment 34.

105. Line 346: Why do you say that current satellites can provide spatially and temporally sparse observations?

106. Line 347: The observations themselves being correlated in time isn't a problem. It is an issue only if the observation errors are correlated.

107. Lines 347 – 350: What is the significance of mentioning "It is important to note… outside the average observation window" and "there is currently no SIT retrieval dataset that provides daily, non-smoothed, and non-time-correlated SIT measurements"?

108. Line 351: Localised dynamics is not the same as having less-smooth fields / more small-scale features. I would interpret "localised dynamics" as having dynamical interactions that are local. This is not easy to quantify, so I suggest avoiding this terminology.

109. Lines 363 – 364: The gradient norm changing by more than an order of magnitude is not a problem. In the extreme case where you converge to the actual minimum, the gradient there is zero, so there is an infinite order of magnitude difference compared to the initial gradient norm.

110. Lines 371 – 373: Do you have a reference for the statement "In deep learning, the high number of degrees of freedom often results in poor-quality gradients"? Also, from a theoretical point of view, how does the background-error term regularise the emulator's noisy gradient?

111. Lines 376 – 377 and Appendix C4: There is also another important test to check that the adjoint has been coded up properly. For $\mathbf{M}^T$ being the adjoint of $\mathbf{M}$, $\langle \mathbf{x}, \mathbf{M}^T \mathbf{y} \rangle$ should be equal to $\langle \mathbf{Mx}, \mathbf{y} \rangle$ (up to machine precision) for any vectors $\mathbf{x}$ and $\mathbf{y}$ of appropriate size, where $\langle \cdot, \cdot \rangle$ is the inner product.

112. Lines 387 – 388: For observations, a more conventional way is to thin the observations before assimilating them. Then the observations would not be regarded as a smooth field, but discrete data points.

113. Line 410 (Equation A1): $N_s$ has not been defined. Please also make the equation and notations consistent with Equations 12 and 13. See also Minor Comment 48.

114. Line 414: It is important to mention at a more prominent place that the outputs of the neural network model are forecast increments instead of the forecast field. Also please explain why this choice is made.

115. Lines 415 – 416: I am under the impression that the mean and standard deviation in Equation 1 are those for the full field, but here you discuss the mean and standard deviation of the forecast increment. Could you please clarify?

116. Algorithms in appendices (in general): Please ensure that all notations in the presented algorithms are defined. While a specialist reader may be able to guess what they are, the presentation of the algorithms needs to be accessible to a general audience.

117. Algorithm A1: It doesn't seem clear to me how the individual lines in the algorithm presented here relate to Equations 4a and 4b, especially since there are normalised and unnormalised fields involved here. A concise, worded presentation of the algorithm would help clarify how it works.

118. Caption of Figure A1: Could you explain what the "Training and validation losses" are? Also, there are two blue lines per panel, so it is not clear what "Blue line corresponds to the training of $g_\theta$" is.

119. Figure A2: Do you know why there remains a bias in the $g_\theta$ forecast?

120. Line 426: Is the "number of state[s] in the ensemble" equivalent to the ensemble size?

121. Line 435: It is good to describe how the EOFs are ordered.

122. Lines 442 – 443: The choice of $m = 7000$ seems quite large to me. In the limit of large $m$ ($m \rightarrow 8871$), you would have $\mathbf{B} = \boldsymbol{\varphi}_m \boldsymbol{\varphi}_m^\mathrm{T} = \mathbf{I}$ (where $\mathbf{I}$ is the identity matrix) due to the orthogonal nature of the EOFs. This means you would recover the 4D-Var-diag case, up to the $\sigma_b^2$ scaling. Usually, people look at only a small number of EOFs in order to reduce the dimensionality of the problem without compromising too much the quality of the results, so I find the choice of $m = 7000$ quite intriguing. What would you like to achieve here? Do you want to get to a close approximation to the 4D-Var-diag case? Figure B1 shows that with $m = 4$ you can already explain about 73% of the variance. How different would this be in the $m = 7000$ case? (See also Minor Comment 100.)

123. Line 446: Is there a reason that you present the algorithm for the 4D-Var-EOF case only?

124. Lines 448 – 449: According to Equation 11 and Algorithm C1, the $\mathbf{H}$ operator is operated on $\mathbf{x}$ space instead of $\mathbf{w}$ space, which is inconsistent with the statement "transformed back into the affine space of the EOFs".

125. Algorithm C1: What are $N_f$ and $\theta$, and why is there a third argument to the function $g_\theta^{N_f}$ when there is only one in the $g_\theta$ in Equation 4a?

126. Algorithm C2:
    a. The algorithm is not about the minimisation itself (which is contained within one line, the one with "L-BFGS"), but about the cycling of the 4D-Var scheme, so the heading may be inappropriate.

b. How is the "loss" in the L-BFGS line related to the cost function $J$ in Algorithm C1?

c. How do you ensure that the analysis (incremented from the background) stays within physical bounds?

d. Would you consider combining Algorithms C1 and C2 and presenting them as one algorithm?

127. Lines 450 – 454: You presented two stopping criteria for the minimisation. Does it stop when either criterion is met or when both are met? Failing to meet those criteria, what is the maximum number of iterations to be run?

128. Line 457: Perhaps "Cost function diagnostics" is a better heading for Appendix C2.

129. Line 459, regarding the words "the seasonality… can also be observed in the cost function": This needs to be elaborated further. It is not clear how the horizontal axis in Figure C1 translates to the timing of the year.

130. Lines 462 – 463: What do you mean by the "adaptive background strategy", and do you mean Appendix E instead of Section E?

131. Figures C1 and C2: Do the figures refer to the 4D-Var-diag or 4D-Var-EOF case, and for which type of observations?

132. Figure C1: Is there a reason to show the evolution of cost function values within a minimisation run? I think you can simply show the analysis $J$, $J_b$ and $J_o$ (the final values at the end of the minimisation) as functions of the cycle time.

133. Figure C2: It is not clear what the "first" and "last" in the caption refer to. This is also inconsistent with the text, which says "the beginning and the end of the DAW" (line 465). I don't think much could be said about the evolution of the cost function gradient throughout the DAW, as you start with zero gradient at the end of the window and add contributions to it as you run the adjoint model backwards in time.

134. Line 485: Do you know why the residual errors increase for $\epsilon < 10^{-7}$?

135. Line 489: Please define what $\mathbf{M}$ is. I think so far you have only described about the (non-linear) emulator and the (linear) adjoint, but not a linearised version of the model that is usually denoted by $\mathbf{M}$.

136. Line 493: Is there a reason to show Equation C8c? It is not used in the test described below.

137. Caption of Figures C3 and C4:

a. The $O(\|\epsilon\|)$ line is not an "evolution", but just an identity line on the graph.

b. In Figure C4, this line is in red, not blue, and what has been referred to as the "Black line" in the caption of Figure C4 should have been the blue line.

138. Line 495: You said earlier (line 488) that you want to test the adjoint, but Equation C9 doesn't have $\mathbf{M}^{\mathrm{T}}$ in it. See also Minor Comment 111.

139. Figure D1:

a. Is this figure for the twin experiment or the CS2SMOS experiment?

b. In the legend, what does "LN" mean?

140. Lines 512 – 513: I am not sure about this sentence. What is the element of (Gaussian) randomness that gives rise to the chi-squared distribution? Also, why would the inflation

of the background-error term depend on the number of observations in the observation-error term?

141. Line 514: Why is $p = 8871 \times N_{\text{obs}} \div 2 =$?
142. Line 517: When you speak about DOF, do you refer to the DOF of the chi-squared distribution, or the dimensionality of the cost function?
143. Lines 519 – 520:
    a. Could you please explain the motivation behind the formulation in Equation E2 and explain why the adaptive inflation factor needs to be multiplied by $\lambda_m$?
    b. What are $J_b^n$ and $J_o^n$?
144. Lines 524 – 525: The terminology "multiplicative addition" is ambiguous: it should either multiply or add, but not both.
145. Lines 526 – 527: According to Figure E1, when the inflation factor is larger than 1 in the constant inflation case, the mRMSE is larger, so I am not sure why you say it yields better results.
146. Lines 528 – 529: For the constant inflation case, using an inflation factor above 1 means emphasising the $J_o$ term more and not the $J_b$ term, as you are assigning more error to the background. Hence, according to Figure E1, it is the emphasising of $J_o$ that undermines the performance of 4D-Var, not the emphasising of $J_b$.
147. Lines 534 – 535: Do you know why you improve the results at beginning of the DAW and worsen them at the end of the DAW when adaptive inflation is used?
148. Line 536: I would interpret "4D-Var is struggling" as having convergence issues in the minimisation, which does not necessarily imply the largeness of the analysis RMSE.

---

## Author Comment (AC1)

**Response To Referee 1**
**for 'Four-dimensional variational data assimilation with a sea-ice thickness emulator'**

Charlotte Durand, Tobias Sebastian Finn, Alban Farchi,
Marc Bocquet, Julien Brajard and Laurent Bertino

April 2025

**RC: Reviewer Comment**; AR: Author Response

**RC:** The manuscript "Four-dimensional variational data assimilation with a sea-ice thickness emulator" by Durand et al presents an evaluation of a 4D-Var data assimilation framework using a data-driven sea ice thickness emulator of the neXtSIM sea ice model. The authors show how, through the emulator's backpropagation capabilities, sea ice thickness observations (both idealized and real) can be assimilated into the emulator using 4D-Var, effectively reducing the emulator bias. I very much enjoyed reading this manuscript and think it's a nice contribution to the literature. In fact, most of the comments I had noted down by the time of the discussion were then answered in the discussion, so thanks! My comments were overall minor, and I think the manuscript is almost ready for publication with a few small edits (see below).

AR: We deeply appreciate the reviewer's thorough and insightful review of our work. In the following, we respond to the comments and raised issues and point to the changes in our manuscript.

**RC: General question regarding methodology**

**RC: Could you just clarify something about the methodology for me. Are the EOFs used for the background covariance static over the course of the DA simulation? My concern early on was the ability of the EOF approach to capture flow-dependent processes, given the strong seasonal cycle of sea ice (you do mention this**

**in the discussion). Is there some expectation that the minimization figures out which EOFs are most important and dynamically weights them (in time) according to w? I would be very interested to see how the approach compares to an Ensemble Kalman Filter (as you also say in the discussion).**

AR: Thank you for your remark. Indeed we investigated the influence of the different EOFs weights and their seasonality although we did not present the results in this paper. They are presented in Durand (2024) in Chapter 7. Regarding the 4D–Var-EOF, we can assess the significance of the different weights associated with the EOFs and evaluate the time dependency of the predominant ones. The results are presented in Fig. 1 of the present document. Firstly, the weights associated with the largest amplitudes correspond to the first EOF coefficients. Secondly, tracking the time evolution of the first coefficient reveals a clear temporal dependency, in line with the annual evolution of the SIT. The second coefficient also exhibits a seasonal behavior, with an increase in amplitude around May. The 4D–Var-EOF approach captures the seasonal variability of the signal, which is expected to be the dominant source of temporal variability. However, the full flow-dependent covariance structure, dependent not only on the season but also on the specific realization of the forecast, cannot be represented by 4D–Var-EOF. Capturing this would require comparison with an ensemble-based method such as the EnKF, which would be interesting but is beyond the scope of this article. Note that a standard EnKF system does not rely on an EOF decomposition. However, if the ensemble were projected onto the same EOF basis, it would be possible to compute time-evolving weights $w$ from the EnKF that vary with both location and time, making a direct comparison impractical.

RC: **Comments**

RC: **L112: I suggest adding a citation to show an example of where observations are typically log-normal. E.g Landy et al 2020.**

AR: Thank you very much for your suggestion, we will add this reference: "as more commonly encountered in sea-ice observations from satellites (Landy et al., 2020)"

RC: **L117 and elsewhere: change "In average," to "On average,"**

AR: Thank you for seeing this, we will correct it and check thoroughly the rest of the paper.

RC: **L129 - L132: Somewhere in this section it might be worth high-**

[Figure]

Figure 1: Upper panel: Values of the EOF weights for different assimilation cycles: the 1st cycle is represented by the blue line, the 10th cycle by the purple line, and the 20th cycle by orange line. Note that the x-axis is on a logarithmic scale to highlight the first weights. Bottom panel: Time evolution of the weights of $\mathbf{x}_a$ associated with the first five EOFs for each cycle. The first coefficient is shown in dark blue, the second in indigo, the third in purple, the fourth in pink, and the fifth in orange.

**lighting a recent paper (Nab et al. 2025) which quantified the effect on DA-derived analysis fields due to varying observational uncertainty on sea ice thickness measurements—Turns out to be quite sensitive.**

AR: Thank you for this paper which we were not aware of. Indeed, this is worth mentioning. We will change the text in L132 to : "Note that Nab et al. (2025) showed that modifying SIT observation uncertainties introduces significant sensitivities during SIT assimilation."

**RC: Figures 3 and 7: Missing text in all labels**

AR: Thank you for noticing this, we will be careful to check if in the PDF the text is lisible.

**RC: L258: Doesn't the RMSE in Fig 5 peak in July? I guess the bias error peaks just before May and then rises again in December? Maybe changing L258 to "from Fig. 5 top" to make it clear which panel in Fig 5 we are looking at**

AR: Yes thank you for noticing that, the RMSE peak is indeed rather in July, we will change the sentence to "The analysis from Fig.5(top) reveals a strong seasonality in results, with the RMSE peaking in July."

**RC: L306-310 : Can you borrow some info from data-driven NWP models which retain sharpness by augmenting loss function**

AR: Yes, indeed! For the emulator, we are exploring alternative loss functions to better preserve sharpness. However, these often result in increased RMSEs, so this remains a work in progress and is beyond the scope of the current paper.

**RC: L350 : Might be worth highlighting here that there are ongoing developments in this space. For example Chen et al and Gregory et al both show ML-based approaches for deriving complete daily sea ice and ocean fields from satellite altimetry at 5 km grid resolution. Both of these approaches model the spatio-temporal covariance of daily fields, rather than simply averaging through time. Although these studies show sea ice freeboard, it is conceivable that daily sea ice thickness observations are on the horizon.**

[Figure]

Figure 2: Evolution of the number of L-BFGS iterations as a function of the truncation index.

AR: Thank you for suggesting those references, we will add the sentence: "Ongoing works are proposing ML-based approaches to derive complete daily sea ice freeboard fields from satellite altimetry at fine spatial resolution (5 km), by modeling the spatio-temporal covariance of daily fields rather than relying solely on temporal averaging (Gregory et al., 2024; Chen et al., 2024)."

RC: **L400: I thought neXtSIM-F was initialized through nudging and not EnKF (L274/275)?**

AR: Thank you for seeing this error, neXtSIM-F is indeed initialized with nudging operationaly, we will correct the sentence: "With limited resources, such as emulating only sea-ice thickness and assimilating CS2SMOS observations, the developed 4D–Var system performs comparably to the operational neXtSIM-F system"

RC: **Appendix B: Can you quantify the time change in the 4D-var minimization when increasing the truncation index m? For example, on L324 you say it's 155 seconds for m=7000. What is the time if m is halved to 3500? I guess I'm wondering what is the cost-accuracy tradeoff.**

AR: We show in Fig. 2 of the present document the number of L-BFGS iterations as a function of the truncation index. The execution time to run the 4D–Var is proportional to the number of iterations. As you can see, the increase of the number of iterations is quite linear with the truncation index. When using this method in operations, the choice of the truncation index would be an important factor for the computational efficiency. Yet, to obtain the best RMSE values, based on our experiments, choosing a value above $m = 5000$ seems preferable.

**References**

Chen, W., Mahmood, A., Tsamados, M., and Takao, S. (2024). Deep random features for scalable interpolation of spatiotemporal data.

Durand, C. (2024). Deep learning, data assimilation and sea-ice dynamics.

Gregory, W., MacEachern, R., Takao, S., Lawrence, I. R., Nab, C., Deisenroth, M. P., and Tsamados, M. (2024). Scalable interpolation of satellite altimetry data with probabilistic machine learning. *Nature Communications*, 15(1).

Landy, J. C., Petty, A. A., Tsamados, M., and Stroeve, J. C. (2020). Sea ice roughness overlooked as a key source of uncertainty in cryosat-2 ice freeboard retrievals. *Journal of Geophysical Research: Oceans*, 125(5).

Nab, C., Mignac, D., Landy, J., Martin, M., Stroeve, J., and Tsamados, M. (2025). Sensitivity to sea ice thickness parameters in a coupled ice-ocean data assimilation system. *Journal of Advances in Modeling Earth Systems*, 17(3).